# Elevated ubiquitin phosphorylation by PINK1 contributes to proteasomal impairment and promotes neurodegeneration

Cong Chen[1†], Tong-Yao Gao[1†], Hua-Wei Yi[2], Yi Zhang[3], Tong Wang[1], Zhi-Ling Lou[1], Tao-Feng Wei[1], Yun-Bi Lu[1], Tingting Li[3], Chun Tang[4*], Wei-Ping Zhang[1,5*]

[1]Department of Pharmacology, Zhejiang University School of Medicine, Hangzhou, China; [2]The First People's Hospital of Jingzhou, First Affiliated Hospital of Yangtze University, Jingzhou, China; [3]Department of Biomedical Informatics, School of Basic Medical Sciences, Peking University Health Science Center, Beijing, China; [4]Beijing National Laboratory for Molecular Sciences, College of Chemistry and Molecular Engineering, Center for Quantitate Biology, Center for Life Science, Academy for Advanced Interdisciplinary Studies, Peking University, Beijing, China; [5]Zhejiang Key Laboratory of Precision Psychiatry, Hangzhou, China

*For correspondence:
Tang_Chun@pku.edu.cn (CT);
weiping601@zju.edu.cn (W-PZ)

[†]These authors contributed equally to this work

## eLife Assessment

This study provides **important** insights into the role of polyUbiquitination in neurodegenerative diseases, elucidating how pUb promotes neurodegeneration by affecting proteasomal function. The findings not only offer a new perspective on the pathophysiology of neurodegenerative diseases but also provide potential targets for developing new therapeutic strategies. The results provide **solid** evidence to support the conclusions.

**Abstract** Ubiquitin (Ub), a central regulator of protein turnover, can be phosphorylated by PINK1 (PTEN-induced putative kinase 1) to generate S65-phosphorylated ubiquitin (pUb). Elevated pUb levels have been observed in aged human brains and in Parkinson's disease, but the mechanistic link between pUb elevation and neurodegeneration remains unclear. Here, we demonstrate that pUb elevation is a common feature under neurodegenerative conditions, including Alzheimer's disease, aging, and ischemic injury. We show that impaired proteasomal activity leads to the accumulation of sPINK1, the cytosolic form of PINK1 that is normally proteasome-degraded rapidly. This accumulation increases ubiquitin phosphorylation, which then inhibits ubiquitin-dependent proteasomal activity by interfering with both ubiquitin chain elongation and proteasome-substrate interactions. Specific expression of sPINK1 in mouse hippocampal neurons induced progressive pUb accumulation, accompanied by protein aggregation, proteostasis disruption, neuronal injury, neuroinflammation, and cognitive decline. Conversely, Pink1 knockout mitigated protein aggregation in both mouse brains and HEK293 cells. Furthermore, the detrimental effects of sPINK1 could be counteracted by co-expressing Ub/S65A phospho-null mutant but exacerbated by over-expressing Ub/S65E phospho-mimic mutant. Together, these findings reveal that pUb elevation, triggered by reduced proteasomal activity, inhibits proteasomal activity and forms a feedforward loop that drives progressive neurodegeneration.

## Introduction

Neurodegeneration, a hallmark of aging and neurodegenerative diseases, is characterized by irreversible loss of neurons, ultimately leading to cognitive and motor impairments (*Cookson et al., 2023*). A common feature of neurodegeneration is the accumulation of ubiquitinated proteins, often resulting from the impairment of protein degradation machineries, particularly the ubiquitin-proteasome system (UPS) (*Davidson and Pickering, 2023*; *Kinger et al., 2024*; *McDade et al., 2024*). The UPS is essential for maintaining cellular proteostasis, as it not only eliminates misfolded and toxic proteins but also dynamically regulates neuronal structure and function through controlled protein degradation (*Costa et al., 2019*; *Krishna et al., 2022*; *Soykan et al., 2021*). Substrate proteins targeted for proteasomal degradation are typically conjugated with K48-linked polyubiquitin chains, which interact non-covalently with ubiquitin receptors in the proteasome (*Komander and Rape, 2012*; *Pohl and Dikic, 2019*; *Swatek and Komander, 2016*). However, UPS activity can be compromised in neurodegeneration in multiple ways, including proteasome modifications, reduced ATP level, direct inhibition by amyloid fibrils, and oxidative stress (*Kinger et al., 2024*; *McDade et al., 2024*; *Tseng et al., 2008*; *Cookson et al., 2023*; *Wojcik and Di Napoli, 2004*). While these factors have been extensively studied in order to preserve proteasomal function, the dynamic changes of ubiquitin have been largely overlooked in the context of neurodegeneration.

Ubiquitin itself undergoes post-translational modifications that can alter its structure and function (*Herhaus and Dikic, 2015*). One such modification is phosphorylation at residue S65, installed by the kinase PINK1 (*Koyano et al., 2014*). PINK1 can exist in two forms: the full-length PINK1 localized to the mitochondrial membrane (*Okatsu et al., 2015*; *Okatsu et al., 2013*), and sPINK1, a cytosolic fragment cleaved from full-length PINK1 (*Takatori et al., 2008*; *Yamano and Youle, 2013*). Full-length PINK1 is activated in response to mitochondrial damages, subsequently phosphorylating ubiquitin and Parkin to initiate mitophagy—a neuroprotective process for the selective removal of damaged mitochondria (*Koyano et al., 2014*; *Lazarou et al., 2015*; *Okatsu et al., 2015*; *Cookson et al., 2023*). However, severe and persistent mitochondrial stress can also lead to mitophagy failure and sustained elevations of S65-pUb levels (*Chin et al., 2023*; *Pollock et al., 2024*). Given the prevalence of mitochondrial dysfunction under neurodegenerative conditions (*Cookson et al., 2023*), pUb accumulation may result from the activation of full-length PINK1 but without effective mitophagy.

Alternatively, elevated pUb levels may arise from inhibited proteasomal degradation. The cytosolic sPINK1, processed from the full-length PINK1 by mitochondrial proteases, is normally rapidly degraded by the proteasome via the N-end rule pathway (*Takatori et al., 2008*; *Yamano and Youle, 2013*). As the UPS function is often compromised in neurodegeneration (*Cookson et al., 2023*), sPINK1 may accumulate and phosphorylate ubiquitin, contributing to the increased pUb levels.

Elevated levels of pUb have been observed in the aged human brain and in Parkinson's disease (*Fiesel et al., 2015*; *Hou et al., 2018*; *Shiba-Fukushima et al., 2017*). While pUb elevation has been proposed as a potential biomarker of neurodegenerative diseases (*Fiesel et al., 2015*; *Hou et al., 2018*), its functional role remains poorly understood. In vitro studies have suggested that phosphorylation alters ubiquitin's structural dynamics and may affect its function (*Dong et al., 2017*; *Tang and Zhang, 2020*; *Wauer et al., 2015a*). Moreover, PINK1 phosphorylation has been shown to inhibit ubiquitin chain elongation (*Wauer et al., 2015b*), and the phosphomimic mutant Ub/S65E has been shown to inhibit protein turnover and reduce cell viability under stress conditions in yeast (*Swaney et al., 2015*). As such, elevated pUb levels may actively contribute to the progression of neurodegenerative diseases.

In the current study, we demonstrate that pUb levels are elevated in various neurodegenerative conditions, including Alzheimer's disease (AD), aging, and ischemic injury. Furthermore, we reveal that elevated pUb levels inhibit proteasomal activity, leading to protein aggregation and neuronal damages. Upon investigating the underlying mechanism at molecular, cellular, and animal levels, we have uncovered a general pathogenic pathway involving pUb that operates across a wide spectrum of neurodegenerative disorders.

## Results

### Elevated pUb levels are a pervasive feature of neurodegeneration

Elevated pUb levels have been observed in the brains of individuals with Parkinson's disease (PD) (*Fiesel et al., 2015*; *Hou et al., 2018*; *Shiba-Fukushima et al., 2017*). In the current study, we extend this observation to Alzheimer's disease (AD), the most prevalent neurodegenerative dementia. We found a marked elevation of both PINK1 and pUb in brain samples from AD patients in the cingulate gyrus brain regions with Aβ plaques, when compared to age- and sex-matched controls (*Figure 1A and B*; *Figure 1—source data 1*). This finding was corroborated in the APP/PS1 mouse model of AD, where increased PINK1 and pUb levels could be detected in neocortex of mouse brains with Aβ compared to wild-type mice (*Figure 1C and D*). On the other hand, PINK1 and pUb levels in neocortex of wild-type and Pink1-knockout mice (Pink1-/-) differ little (*Figure 1C and D*), which can be attributed to the low expression of PINK1 in normal mouse brains under physiological conditions.

Beyond the specific neurodegenerative diseases, pUb elevation also appears to be associated with aging process (*Fiesel et al., 2015*; *Hou et al., 2018*), a special pathological condition that is often accompanied by gradual neurodegeneration. The difference in PINK1 expression between young and aged mouse cortex was not distinguishable due to the low expression level and probably also due to specificity of PINK1 antibody used. However, we observed a significant increase in neuronal pUb levels in aged wild-type mouse compared to young mouse in neocortex of the brains (*Figure 1E and F*). As a control, pUb levels in Pink1-/- mice remained unchanged with age and were notably lower than those in aged wild-type mice (*Figure 1E and F*). As such, although the pUb antibody may not be highly specific, the observed increase in immunofluorescence and protein band intensities, when comparing across different Pink1 genetic background, can only be attributed to elevated pUb levels during aging.

We further showed that acute neurodegenerative conditions, such as cerebral ischaemia, are also associated with elevated pUb levels. Using mouse middle cerebral artery occlusion (MCAO) model, we observed a marked increase in both PINK1 and pUb levels in the ischemic core of mouse brains compared to the contralateral cortex (*Figure 1G*, *Figure 1—figure supplement 1*). To further explore the relationship between pUb and ischemic stress, we subjected HEK293 cells to oxygen-glucose deprivation (OGD), a cellular model mimicking ischemic conditions. OGD followed by reperfusion caused a time-dependent increase in PINK1, sPINK1, and pUb levels (*Figure 1H*). Additionally, we observed an increase in protein aggregation, as evidenced by the accumulation of ubiquitin in the insoluble protein fraction (*Figure 1I*).

Taken together, these findings demonstrate that elevated pUb levels are a common feature across a wide spectrum of neurodegenerative conditions, both chronic and acute. While our observation supports the notion of pUb as a biomarker, the diverse functions of ubiquitin suggest that elevated pUb levels may actively contribute to the pathogenesis of neurodegeneration.

### Ubiquitin phosphorylation by sPINK1 impairs proteasomal function in HEK293 cells

Elevated pUb levels can result from either an elevated level of full-length PINK1 or sPINK1. PINK1 increase can occur upon mitochondrial injuries with the treatment of CCCP and O/A, and sPINK1 increase can occur upon proteasomal inhibition with the treatment of MG132 (*Figure 2A*). The treatment of MG132 caused a concentration- and time-dependent increase in sPINK1 and pUb levels in wild-type HEK293 cells, with sPINK1 levels plateaued at 6 hr (*Figure 2B–E*). In contrast, PINK1 and pUb levels remained low in PINK1-knockout HEK293 cells upon the treatment of MG132 (*Figure 2B–E*).

While PINK1 activation typically involves its dimerization at mitochondrial outer membrane (*Gan et al., 2022*; *Okatsu et al., 2013*), we investigated whether cytoplasmic sPINK1 retains activity of phosphorylating ubiquitin. We transiently transfected HEK293 cells with several PINK1 constructs: full-length PINK1, a cytoplasmic variant of sPINK1 that is more resistant to UPS degradation (sPINK1*, PINK1/F101M/L102-L581, differing from the native sequence by the N-terminal Met instead of Phe), a kinase-dead version of this stable variant (sPINK1*-KD, with K219A/D362A/D384A mutations introduced) (*Beilina et al., 2005*), and Ub^GG-sPINK1, a short-lived version of native-sequence sPINK1 with a ubiquitin appended at the N-terminus (*Yamano and Youle, 2013*). Western blot analysis confirmed the expression of PINK1, sPINK1*, and sPINK1*-KD following transfection, while Ub^GG-sPINK1 was undetectable due to its rapid degradation (*Figure 2F*). The analysis also showed elevated pUb levels

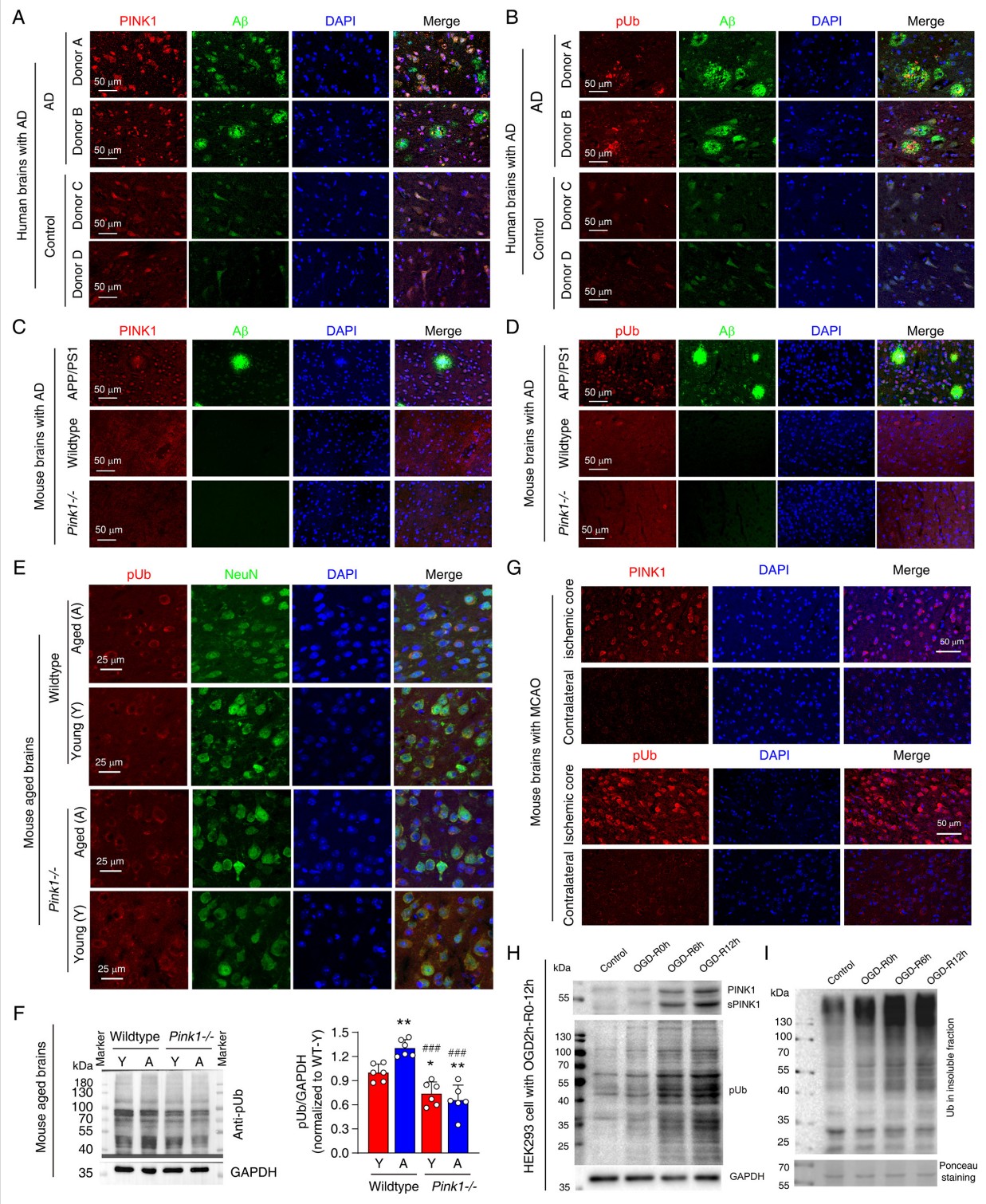

**Figure 1.** Elevated phosphorylated ubiquitin (pUb) levels are widespread across neurodegenerative conditions. (**A**, **B**) Double immunofluorescence staining showing the distribution of PTEN-induced kinase 1 (PINK1) and Aβ (**A**), and pUb and Aβ (**B**), within the cingulate gyrus brain region of Alzheimer's disease (AD) patients compared to similar brain regions from age-matched controls. Detailed donor information is provided in *Figure 1—source data 1*. (**C**, **D**) Double immunofluorescence staining of PINK1 and Aβ (**C**), and pUb and Aβ (**D**) in the brains of wild-type, Pink1-/-, and APP/PS1 transgenic mice. The images were taken for the neocortex of APP/PS1, wild-type, and Pink1-/- mice. (**E**) Double immunofluorescence staining for pUb and the neuronal marker NeuN in the neocortex of young and aged brains of both wild-type and Pink1-/- mice. The images were taken for the layer III-IV of neocortex. (**F**) Western blot analysis of pUb levels in the cortex of young (Y) and aged (A) wild-type and Pink1-/- mouse brains, quantitatively

*Figure 1 continued on next page*

*Figure 1 continued*

comparing protein levels across ages and genotypes. N=6, *p<0.05, **p<0.01 for comparisons with young wild-type mice; ###p<0.001 for comparisons with aged wild-type mice, one-way ANOVA. (**G**) Immunofluorescence staining for PINK1 and pUb in the contralateral and penumbra of mouse brains subjected to middle cerebral artery occlusion (MCAO) for 2 hr, followed by 24 hr of reperfusion. Locations of the analyzed brain regions are shown in *Figure 1—figure supplement 1*. (**H**) Western blot analysis of PINK1 and pUb levels in HEK293 cells subjected to oxygen-glucose deprivation (OGD) for 2 hr, followed by reperfusion at 0, 6, and 12 hr. (**I**) Western blot analysis of ubiquitin in the insoluble fraction of HEK293 cells post 2 hr OGD and subsequent 0, 6, 12 hr of reperfusion.

The online version of this article includes the following source data and figure supplement(s) for figure 1:

**Source data 1.** Word file of a table containing the clinical and pathological characteristics of brain donors.

**Source data 2.** PDF file containing original western blots for *Figure 1F*, indicating the relevant bands and treatments.

**Source data 3.** Original files for western blot analysis displayed in *Figure 1F*.

**Figure supplement 1.** Nissl staining brain section from a mouse with middle cerebral artery occlusion (MCAO) to show the brain regions.

upon the transfection of full-length PINK1 and sPINK1*, with the latter more pronounced. In contrast, transfection with sPINK1*-KD or Ub$^{GG}$-sPINK1 had little effect on pUb levels (*Figure 2F*).

Furthermore, immunofluorescence staining revealed a diffuse cytoplasmic ubiquitin distribution in non-transfected or eGFP-transfected control cells. In contrast, ubiquitin-positive puncta were observed following MG132 treatment or upon the transfection of sPINK1* (*Figure 2G*). Weak ubiquitin-positive puncta were also visible in Ub$^{GG}$-sPINK1-transfected cells (*Figure 3G*, fifth row, indicated by arrows), likely due to a slight elevation in sPINK1 level. Furthermore, Western blot analysis showed that MG132 treatment and sPINK1* transfection (*Figure 2H*), but not sPINK1*-KD or Ub$^{GG}$-sPINK1 transfection, increased ubiquitin levels in the insoluble protein fraction. These findings suggest a positive correlation between PINK1 and pUb levels and the aggregation of ubiquitinated proteins, which likely results from an impairment of proteasomal activity, akin to the effect of MG132.

To confirm the inhibitory effect of pUb on proteasomal activity, we transfected cells with Ub-R-GFP, a model substrate for the proteasome (*Dantuma et al., 2000*; *Samant et al., 2018*). GFP was rapidly degraded by functional proteasomes but accumulated upon the administration of MG132. Transfection with sPINK1* also caused GFP accumulation, but the transfection of sPINK1*-KD did not (*Figure 2I*). Thus, sPINK1* transfection leads to increased pUb levels and impairs proteasomal degradation.

Elevated pUb levels may also lead to protein aggregation by inhibiting autophagic degradation. To assess the involvement of this pathway, we treated HEK293 cells with puromycin to increase ubiquitinated protein level, which can be degraded through both autophagic and UPS pathways (*Figure 2I*). The ubiquitin signal was significantly lower in the sPINK1*-transfected cells compared to neighboring non-transfected cells (*Figure 2J*), indicating that sPINK1* promotes the degradation of ubiquitinated proteins. However, when we blocked autophagic degradation using bafilomycin A1 (BALA) (*Mauvezin and Neufeld, 2015*), ubiquitin signal was higher in the sPINK1*-transfected cells compared to non-transfected cells (*Figure 2J*), indicating sPINK1* inhibits the degradation of ubiquitinated protein. Taken together, the findings demonstrate that while sPINK1* can enhance autophagic degradation of ubiquitinated proteins, potentially through p62 phosphorylation as previously reported (*Gao et al., 2016*), elevated PINK1 and pUb levels cause an inhibition of proteasomal degradation.

## pUb impairs proteasomal activity through both covalent and noncovalent mechanisms

To elucidate how elevated pUb levels inhibit UPS activity, we first assessed ubiquitin chain formation in vitro. While the unmodified ubiquitin efficiently formed di-, tri-, and higher-order chains catalyzed by E1 and E2 enzymes, pUb predominantly formed di-Ub chains (*Figure 3A*), consistent with the previous report (*Wauer et al., 2015b*). The introduction of S65A (phospho-null) and S65E (phospho-mimetic) ubiquitin mutants further confirmed the impact of phosphorylation—Ub/S65A mutation did not affect chain elongation, producing a polyUb ladder similar to unmodified ubiquitin, while Ub/S65E mutation only formed di-Ub (*Figure 3A*).

Furthermore, we examined ubiquitin chain formation in HEK293 cells by expressing FLAG-Ub/48K, a ubiquitin mutant that can only form K48-linked ubiquitin chains. Upon proteasomal inhibition with MG132, K48-linked chains were observed in wild-type and PINK1-/- cells, but were absent in cells

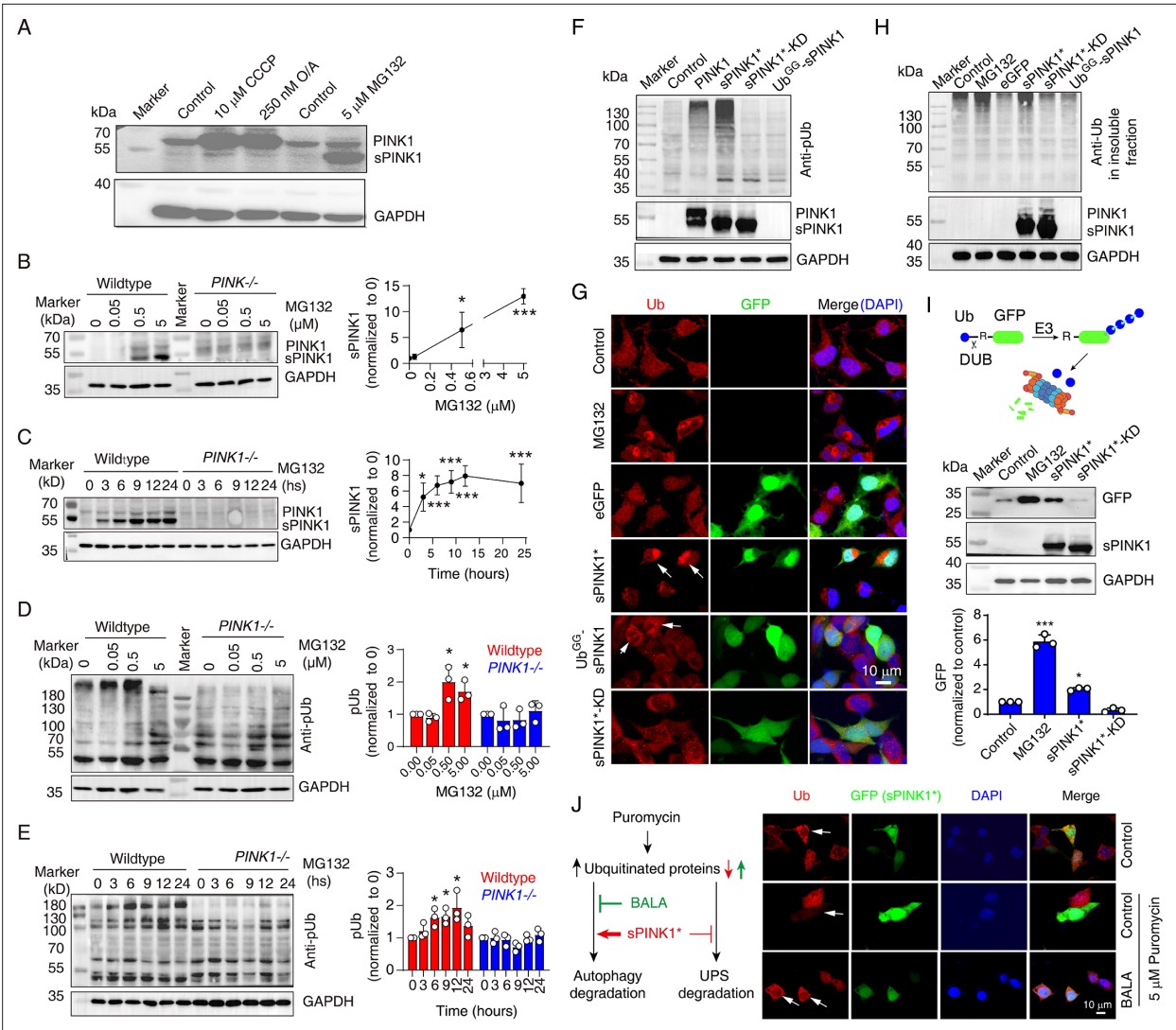

**Figure 2.** Ubiquitin phosphorylation by sPINK1 affects proteasomal activity in HEK293 cells. (**A**) Representative Western blot showing the levels of PINK1 following the administration of CCCP, O/A, or MG132. The CCCP was treated for 12 hr, O/A was treated for 2 hr, and the MG132 was treated for 8 hr. (**B**) Western blot analysis showing the concentration-dependent effect of MG132 (0–5 μM) over an 8 hr period on PINK1 level. N=3; *p<0.05, ***p<0.001, compared to 0 μM MG132, one-way ANOVA. (**C**) Western blot analysis showing the time-dependent effect of 5 μM MG132 on PINK1 levels in 0–24 hr. N=4; *p<0.05, ***p<0.001, compared to 0 hr, one-way ANOVA. (**D**) Western blot analysis of phosphorylated ubiquitin (pUb) levels under a concentration gradient of MG132 (0–5 μM) for 8 hr. N=3; *p<0.05, compared to 0 μM MG132, one-way ANOVA. (**E**) Western blot analysis of pUb levels over a time course of 0–24 hr with 5 μM MG132 treatment. N=3; *p<0.05, compared to 0 hr, one-way ANOVA. (**F**) Representative Western blot image showing the levels of pUb and PINK1 at 24 hr post-transfection with different PINK1 constructs: sPINK1* (PINK1/F101M/L102-L581), sPINK1*-KD (kinase-dead sPINK1* with additional K219A/D362A/D384A mutations), and Ub^GG-sPINK1 (a short-lived version of native sPINK1 with an appended N-terminal Ub). (**G**) Representative immunofluorescence images of ubiquitin staining in cells treated with 5 μM MG132 for 8 hr or transfected with different sPINK1 constructs, highlighting differences in ubiquitin localization and aggregation. The white arrows indicate positively transfected cells. (**H**) Representative Western blot image of ubiquitin in the insoluble protein fraction of cells. The cells were collected at 8 hr after treatment with 5 μM MG132, or collected at 24 hr after transfection with different sPINK1 constructs. (**I**) Western blot analysis showing GFP degradation in HEK293 cells transfected with Ub-R-GFP. Cells were harvested at 8 hr after 5 μM MG132 treatment, or 24 hr following sPINK1* or sPINK1*-KD transfection. N=3; *p<0.05, ***p<0.001, compared to the control, one-way ANOVA. (**J**) Immunofluorescence staining of ubiquitin illustrating how sPINK1 overexpression impacts on proteasomal and autophagic degradation. Puromycin blocks protein synthesis at the translation elongation stage, leading to the production of truncated proteins; BALA (bafilomycin A1, a v-ATPase inhibitor) blocks the degradation of autophagic cargo by inhibiting autophagosome-lysosome fusion, as illustrated in the left panel. Puromycin (5 μg/ml) was applied for 2 hr before harvesting with or without the treatment of 0.1 μM BALA. The white arrows indicate positively transfected cells.

The online version of this article includes the following source data for figure 2:

**Source data 1.** PDF file containing original western blots for *Figure 2B*, indicating the relevant bands and treatments.

*Figure 2 continued on next page*

*Figure 2 continued*

**Source data 2.** Original files for western blot analysis displayed in *Figure 2B*.

**Source data 3.** PDF file containing original western blots for *Figure 2C*, indicating the relevant bands and treatments.

**Source data 4.** Original files for western blot analysis displayed in *Figure 2C*.

**Source data 5.** PDF file containing original western blots for *Figure 2D*, indicating the relevant bands and treatments.

**Source data 6.** Original files for western blot analysis displayed in *Figure 2D*.

**Source data 7.** PDF file containing original western blots for *Figure 2E*, indicating the relevant bands and treatments.

**Source data 8.** Original files for western blot analysis displayed in *Figure 2E*.

**Source data 9.** PDF file containing original western blots for *Figure 2I*, indicating the relevant bands and treatments.

**Source data 10.** Original files for western blot analysis displayed in *Figure 2I*.

over-expressing full-length PINK1 or sPINK1* (*Figure 3B*). This further confirms the inhibitory effect of S65 phosphorylation on ubiquitin chain elongation.

Since PINK1 can phosphorylate both ubiquitin monomers and polyUb chains (*Shiba-Fukushima et al., 2014*), we next investigated how polyUb phosphorylation impacts proteasomal degradation. Using an in vitro degradation assay using purified proteasomes, we found that the degradation of K48-linked pUb-chain-modified GFP (pK48-polyUb-GFP) was slower compared to its unmodified counterpart (K48-polyUb-GFP) (*Figure 3C*, *Figure 3—figure supplements 1 and 2*). Additionally, we employed total internal reflection fluorescence (TIRF) microscopy, following an established protocol (*Lu et al., 2015*), to assess interactions between ubiquitinated substrate proteins and immobilized proteasomes. The assay revealed fewer GFP puncta for pK48-polyUb-GFP compared to K48-polyUb-GFP (*Figure 3D, E*, *Figure 3—figure supplement 3*). Furthermore, the GFP puncta for pK48-polyUb-GFP dissociated more rapidly from the proteasomes, reflecting a shorter dwell time of the proteasome-substrate complex (*Figure 3F–I*, *Figure 3—figure supplement 4*). Thus, phosphorylation impairs the noncovalent interactions between polyUb chains conjugated to substrate proteins and the ubiquitin receptors residing in the proteasome.

Together, these findings demonstrate that elevated pUb levels disrupt proteasomal function through two distinct mechanisms: impairing covalent ubiquitin chain elongation and substrate ubiquitination, as well as weakening non-covalent interactions between ubiquitinated substrates and the proteasome. Both mechanisms collectively contribute to the inhibition of ubiquitin-dependent proteasomal activity.

## Pink1 knockout mitigates protein aggregation under neurodegenerative conditions

Having established the causative link between elevated pUb levels and proteasomal inhibition, we further explored the relationship with protein aggregation and whether Pink1 knockout mitigates this process. We treated wild-type and PINK1-/- HEK293 cells with MG132, which increased insoluble ubiquitin levels in both cell types in a concentration- and time-dependent manner (*Figure 4A and B*). However, insoluble ubiquitin levels accumulated significantly slower in PINK1-/- cells, though eventually reaching levels comparable to those in wild-type cells (*Figure 4B*). This not only confirms that PINK1/pUb operates within the same pathway as MG132—the proteasome—but also demonstrates that PINK1 deficiency delays the accumulation of protein aggregates. MG132 treatment also elevated the levels LC3-II, a marker of autophagy, in wild-type cells, indicating either autophagy activation or reduced LC3 degradation in response to proteasomal inhibition (*Figure 4C and D*). However, LC3-II levels did not increase in PINK1-/- cells following MG132 treatment (*Figure 4C and D*). This confirms the critical role of PINK1 in the activation of autophagy (*Figure 2J*).

Furthermore, Pink1 knockout mitigated protein aggregation in animal models. In young wild-type mouse brains, ubiquitin was evenly distributed in the neurons in neocortex, but formed ubiquitin-positive puncta in aged wild-type mouse brains (*Figure 4E*). Western blot analysis revealed elevated levels of insoluble ubiquitin in the cortex of aged mice compared to young mice (*Figure 4F*), further supporting age-related protein aggregation. In contrast, aged Pink1-/- mice showed no such age-related changes in neuronal ubiquitin distribution or insoluble ubiquitin levels (*Figure 4E and F*). Following MCAO, insoluble ubiquitin levels were significantly elevated in the ipsilateral hemisphere of

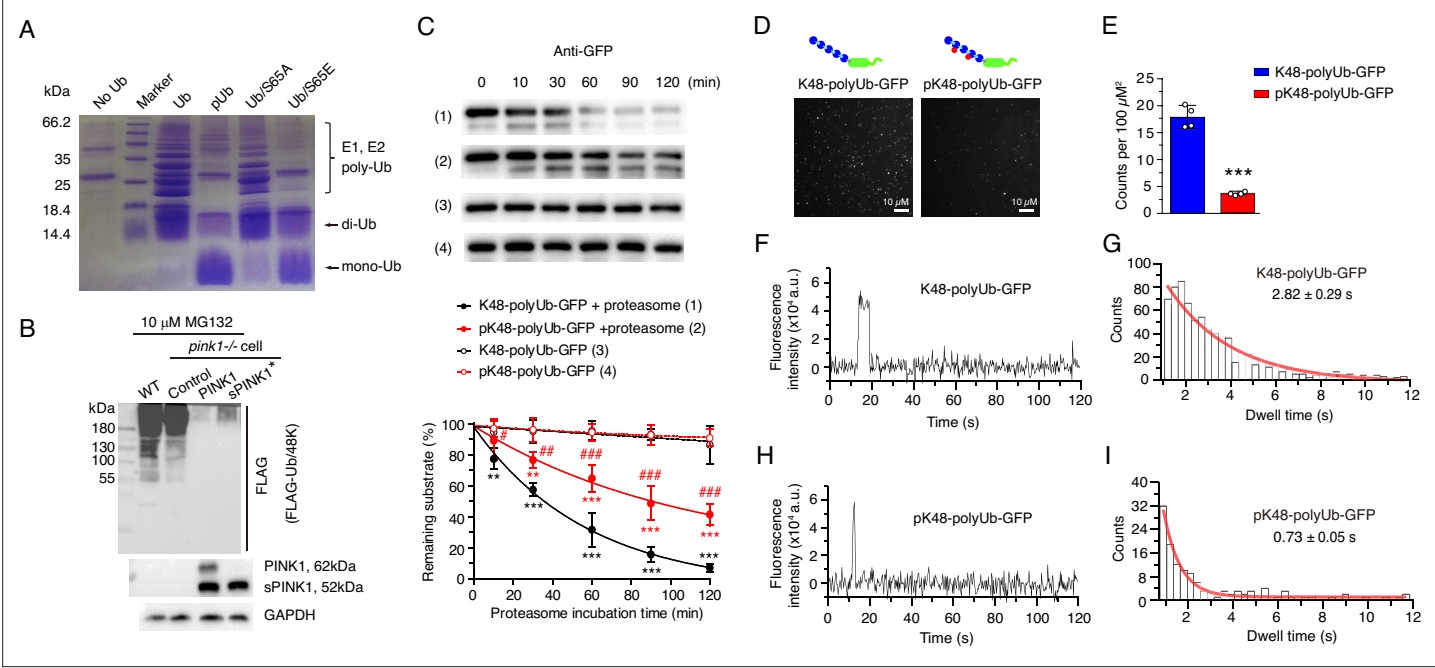

**Figure 3.** Ubiquitin phosphorylation inhibits both ubiquitin chain elongation and ubiquitin proteasome interactions. (**A**) Coomassie blue staining showing ubiquitin chain formation/elongation using different ubiquitin variants as building blocks: wild-type ubiquitin, phosphorylated ubiquitin (pUb), phospho-null Ub/S65A, and phospho-mimic Ub/S65E. (**B**) Western blot showing the formation of K48-linked ubiquitin chain in wild-type (WT) and pink1-/- HEK293 cells without or with PINK1 and sPINK1* transfection. Cells were transfected with FLAG-tagged Ub/48 K to form K48-linked ubiquitin chain only, and 10 µM MG132 was applied at 12 hr post-transfection to prevent the degradation of ubiquitinated substrates. (**C**) Western blot analysis of in vitro proteasomal degradation of GFP modified with K48-linked ubiquitin chains (K48-polyUb-GFP) and GFP modified with phosphorylated K48-linked ubiquitin chains (pK48-polyUb-GFP). N=3; **p<0.01, ***p<0.001 compared to respective controls without added proteasome; #p<0.05, ##p<0.01, ###p<0.001 compared to K48-polyUb-GFP with proteasome, two-way ANOVA. (**D**) Representative TIRF microscopy images showing single-molecule association of K48-polyUb-GFP (left) and pK48-polyUb-GFP (right) to the surface-immobilized proteasomes, visualized as bright puncta. Details of total internal reflection fluorescence (TIRF) single-molecule visualization is shown in *Figure 3—figure supplement 3*. (**E**) Quantitative analysis of puncta density from TIRF images comparing the number of puncta of K48-polyUb-GFP and pK48-polyUb-GFP associated with surface-immobilized proteasomes. N=4; ***p<0.001 compared to K48-polyUb-GFP, using a paired t-test. (**F**, **H**) Representative fluorescence time traces of a single punctum for proteasomal bind K48-polyUb-GFP (**F**) and pK48-polyUb-GFP (**H**). More representative figures of single-molecule fluorescence are shown in *Figure 3—figure supplement 4*. (**G**, **I**) Analysis of GFP fluorescence dwell time for K48-polyUb-GFP (**G**) and pK48-polyUb-GFP (**I**) associated with the proteasome. Data were binned and modeled with a single exponential decay curve (red line).

The online version of this article includes the following source data and figure supplement(s) for figure 3:

**Source data 1.** PDF file containing original western blots for *Figure 3C*, indicating the relevant bands and treatments.

**Source data 2.** Original files for western blot analysis displayed in *Figure 3C*.

**Figure supplement 1.** Preparation of the 26 S proteasome for in vitro degradation of proteins.

**Figure supplement 2.** Preparation of GFP substrate protein modified with ubiquitin (Ub) chain.

**Figure supplement 3.** Visualization of proteasome-associated ubiquitin (Ub)-modified GFP using total internal reflection fluorescence (TIRF).

**Figure supplement 4.** Time traces of polyUb-tagged GFP substrate visualized with total internal reflection fluorescence (TIRF).

wild-type mice compared to sham and contralateral hemisphere, but remained unchanged in Pink1-/- mice (*Figure 4G*).

Protein aggregation is a hallmark of aging and neurodegenerative diseases such as AD (*Hegde et al., 2023*); it arises from perturbed proteostasis and can impair proteasomal activity in return. Acute proteasomal inhibition can also result from ischaemia or pharmacological treatments. Since sPINK1 is rapidly degraded by functional proteasomes, impaired proteasomes can lead to the accumulation of sPINK1 and pUb, leading to further proteasomal inhibition. Together, our results reveal that Pink1 knockout disrupts this cycle, thereby alleviating proteasomal inhibition and mitigating protein aggregation under these conditions, further demonstrating the critical role of pUb in this process.

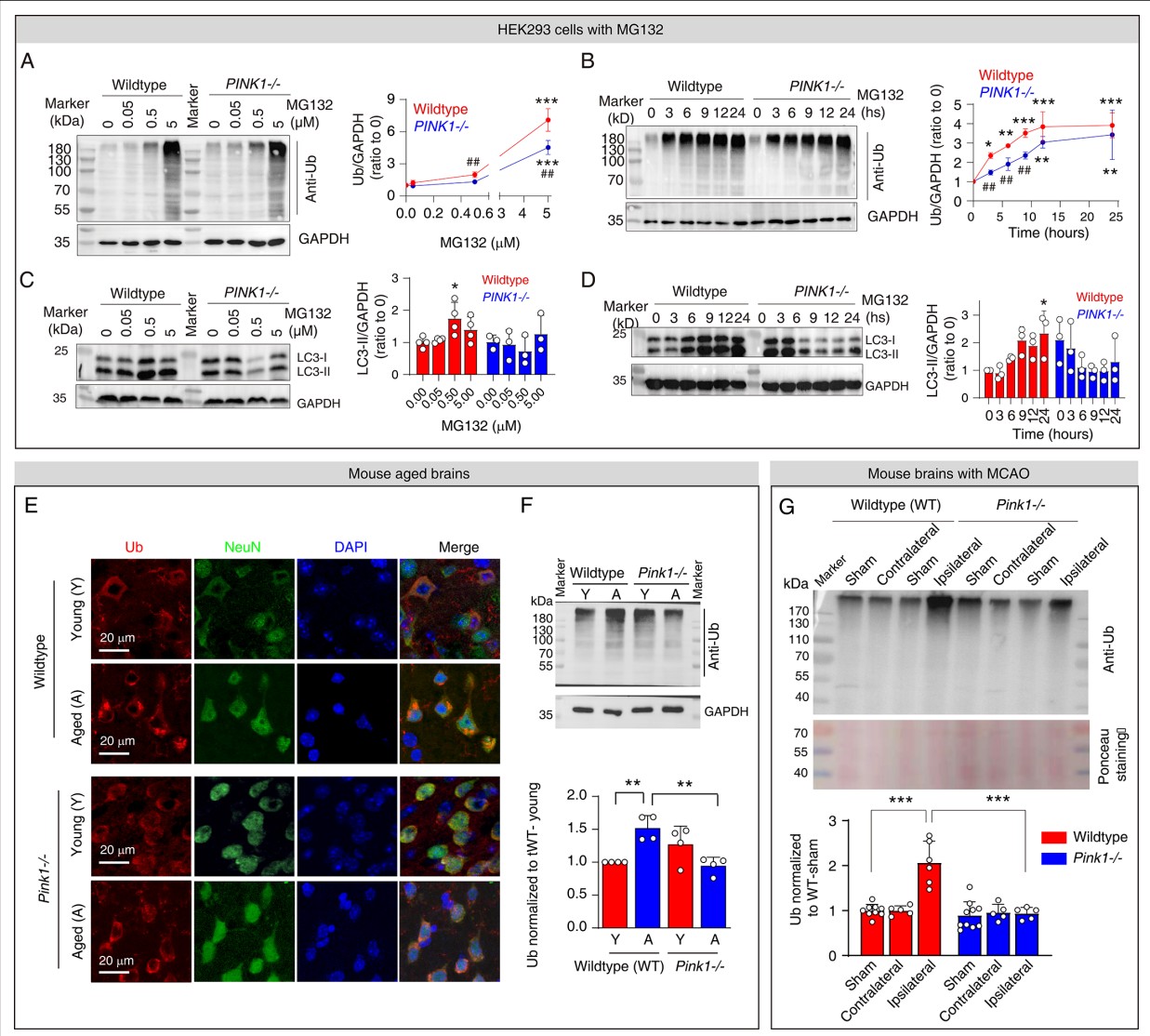

**Figure 4.** Pink1 knockout mitigates protein aggregation upon proteasomal inhibition. (**A**) Western blot analysis of ubiquitin levels in the insoluble protein fraction of HEK293 cells following the treatment with 0–5 μM MG132 for 8 hr. N=5; ***p<0.001 compared to 0 μM MG132; ##p<0.01 compared with wild-type cells, one-way ANOVA. (**B**) Western blot analysis of ubiquitin levels in the insoluble protein fraction of HEK293 cells following the treatment with 5 μM MG132 over 24 hr period. N=3; *p<0.05, **p<0.01, ***p<0.001 compared to 0 μM MG132; ##p<0.01 compared with wild-type cells, one-way ANOVA. (**C**) Western blot analysis of LC3 levels in HEK293 cells following the treatment with 0–5 μM MG132 for 8 hr. N=3–4; *P<0.05 compared to 0 μM MG132, one-way ANOVA. (**D**) Western blot analysis of LC3 levels in HEK293 cells following the treatment with 5 μM MG132 treatment over 24 hr. N=3; *p<0.05 compared to 0 μM MG132, one-way ANOVA. (**E**) Immunofluorescence staining of ubiquitin taken from layer III-IV in the neocortex of brains from young and aged wild-type and Pink1-/- mice. (**F**) Western blot analysis quantifying ubiquitin in the insoluble protein fraction of the cortex from young and aged mouse brains. N=4; **p<0.01, one-way ANOVA. (**G**) Western blot analysis of ubiquitin in the insoluble protein fraction of wild-type and Pink1-/- mouse brains subjected to middle cerebral artery occlusion (MCAO) for 2 hr followed by 24 hr of reperfusion. Comparison includes the sham-operated group (same procedure without occlusion). Locations of the analyzed regions (contralateral and ipsilateral) are indicated in *Figure 1—figure supplement 1*. N=5–9, ***p<0.001, one-way ANOVA.

The online version of this article includes the following source data for figure 4:

**Source data 1.** PDF file containing original western blots for *Figure 4A*, indicating the relevant bands and treatments.

**Source data 2.** Original files for western blot analysis displayed in *Figure 4A*.

**Source data 3.** PDF file containing original western blots for *Figure 4B*, indicating the relevant bands and treatments.

**Source data 4.** Original files for western blot analysis displayed in *Figure 4B*.

**Source data 5.** PDF file containing original western blots for *Figure 4C*, indicating the relevant bands and treatments.

*Figure 4 continued on next page*

*Figure 4 continued*

**Source data 6.** Original files for western blot analysis displayed in *Figure 4C*.

**Source data 7.** PDF file containing original western blots for *Figure 4D*, indicating the relevant bands and treatments.

**Source data 8.** Original files for western blot analysis displayed in *Figure 4D*.

**Source data 9.** PDF file containing original western blots for *Figure 4F*, indicating the relevant bands and treatments.

**Source data 10.** Original files for western blot analysis displayed in *Figure 4F*.

**Source data 11.** PDF file containing original western blots for *Figure 4G*, indicating the relevant bands and treatments.

**Source data 12.** Original files for western blot analysis displayed in *Figure 4G*.

## sPINK1* over-expression causes cumulative proteome-wide changes

Given the critical role of ubiquitin phosphorylation in proteasomal inhibition and protein aggregation, we sought to confirm these effects using phosphomimetic Ub/S65E mutant in SH-SY5Y cell, a neuroblast-link cell line. Overexpression of sPINK1* or Ub/S65E induced the formation of ubiquitin-positive puncta in cells, similar to the effects of MG132 treatment (*Figure 5—figure supplement 1*) and the observations in HEK293 cells (*Figure 2G*).

To extend our findings, we over-expressed sPINK1*, sPINK1* with Ub/S65A (sPINK1*+Ub/S65A), Ub/S65E, or GFP control in mouse hippocampal neurons using AAV2/9. Specific expression was confirmed through GFP co-localization with NeuN fluorescence in the CA1 region (*Figure 5—figure supplement 2*). While no visible changes were observed at 10 days post-transfection, significant neuronal loss, based on NeuN immunofluorescence staining, was evident at 30 and 70 days in mice expressing Ub/S65E, but not in other three groups (*Figure 5—figure supplement 2*). Additionally, Ub/S65E induced glial responses at 30- and 70 days post-transfection, based on GFAP/Iba1 immunofluorescence staining and GFAP/CD11b Western blot bands (*Figure 5—figure supplements 3 and 4*, respectively). In contrast, the sPINK1* and sPINK1*+Ub/S65A groups showed an increase in GFAP levels at 70 days post-transfection with no changes in Iba1 or CD11b levels, suggesting potential neuronal injury (*Figure 5—figure supplements 3 and 4*).

To explore the underlying mechanism for neuronal injury, we conducted proteomic analysis of mouse hippocampus at 30- and 70-days post-transfection. Although the total number of proteins identified was similar between the GFP and sPINK1* groups (*Figure 5A–D*), notable differences were observed. For example, the neuronal injury and neuroinflammation marker HMGB1 was elevated at both time points following sPINK1* overexpression, while the mitochondrial marker Tom20 showed a delayed decrease at 70 days, indicative of mitochondrial loss.

We also conducted gene set enrichment analysis (GSEA) and found a time-dependent effect of sPINK1* overexpression on mitochondrial function. At 30 days, changes in proteins associated with mitophagy, mitochondrial organization, and respiratory electron transport were observed, suggesting an initial injury and compensatory response to mitochondrial stress (*Figure 5E*). By 70 days, these changes expanded to include broader mitochondrial dysfunction and apoptotic processes, ultimately leading to mitochondrial loss and neuronal injury (*Figure 5E*).

Additional gene ontology (GO) analysis revealed extensive proteomic dysregulation upon sPINK1* overexpression (*Figure 5F–H*). At 70 days post-transfection, a large number of proteins were either upregulated or downregulated, indicative of disrupted proteostasis. This can be attributed to proteasomal inhibition or stress response to increased protein aggregates. For example, we observed a significant up-regulation of RNA-binding and RNA-processing proteins. Significantly, many of the up-regulated proteins were associated with impaired synaptic functions and neuronal projections, while down-regulated proteins were linked to myelination, synaptic maturation, and lipid metabolism (*Figure 5F–H*). As these changes were insignificant at 30 days post-transfection (*Figure 5F–H*), the proteomics data indicate a systemic and time-dependent impact upon sPINK1* over-expression on neuronal structure and function, leading to cumulative neuronal injury that mirrors the progressive pathogenesis of neurodegeneration.

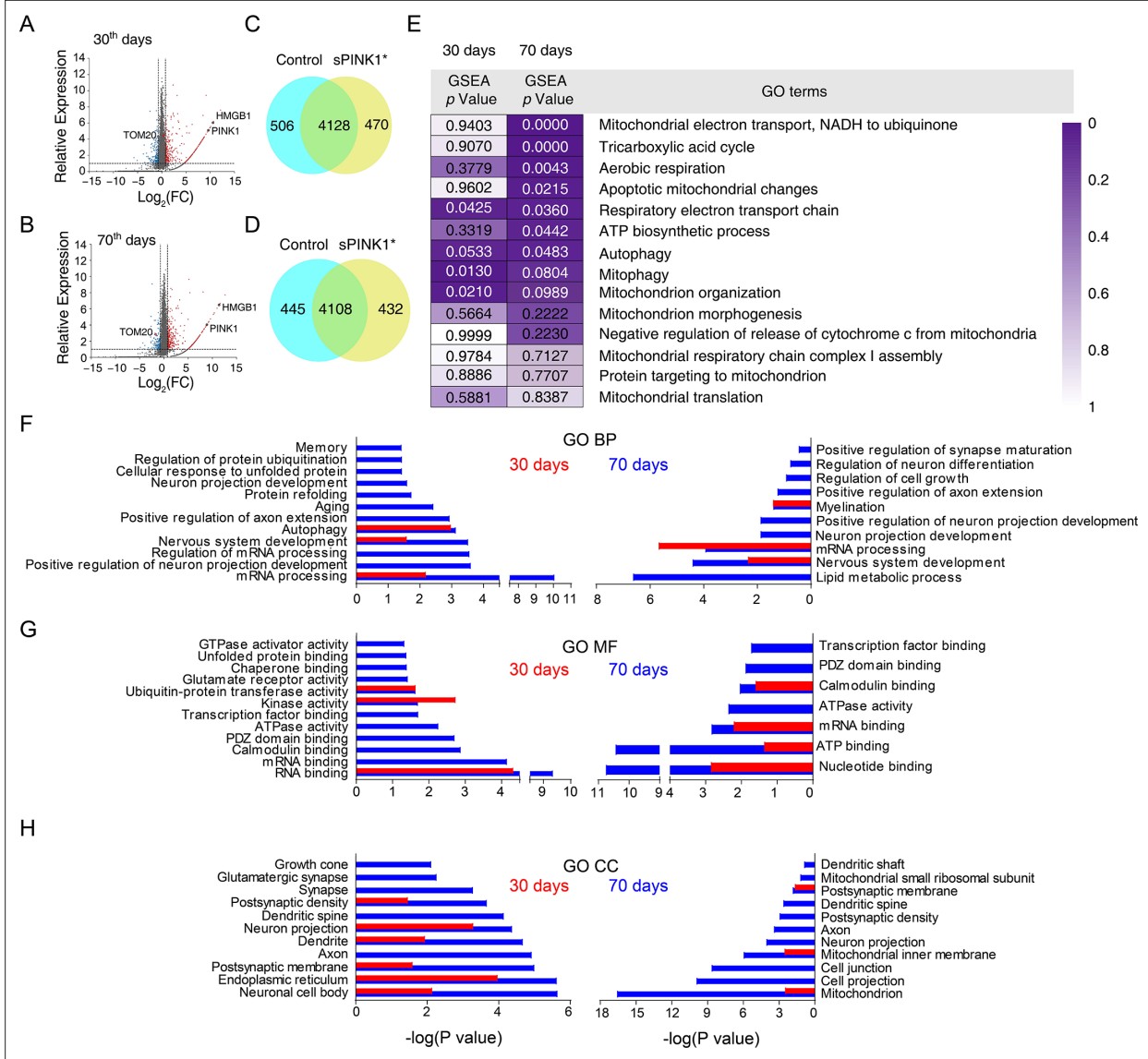

**Figure 5.** Proteomics analysis of the mouse hippocampus at 30-and 70 days post-transfection. (**A** and **B**) Volcano plot of the proteomics data from the mouse brains at 30- and 70 days post-transfection, respectively. (**C** and **D**) Proteomic analysis revealed differential set of proteins upon sPINK1* over-expression. (**E**) The GSEA analysis of mitochondria-related GO terms based on the proteomic data, with the statistical significance values color-coded. (**F**–**H**) Gene ontology (GO) term analysis of the proteomics data for biological process (BP) (**F**), molecular function (MF) (**G**), and cellular component (CC) (**H**), respectively. Red columns denote the data at 30 days and blue columns at 70 days post-transfection. Left panels denote proteins up-regulated by twofold or more, and right panels down-regulated by 50% or more.

The online version of this article includes the following source data and figure supplement(s) for figure 5:

**Source data 1.** Excel file containing original proteomic data for *Figure 5* indicating the relative protein expression levels upon sPINK1 over-expression in mouse hippocampus.

**Figure supplement 1.** Overexpression of sPINK1 induced protein aggregation in SH-SY5Y cells.

**Figure supplement 2.** Representative immunofluorescence images of the hippocampus CA1 regions using anti-adeno-associated virus NeuN antibody.

**Figure supplement 3.** Representative immunofluorescence images of the hippocampus CA1 regions using anti-GFAP and anti-Iba1 antibody.

**Figure supplement 4.** Overexpression of sPINK1* induced gliosis at 70 days post-transfection.

## Elevated pUb levels promote protein aggregation in mouse hippocampal neurons

To investigate the impact of sPINK1* overexpression on protein homeostasis, we measured the levels of PINK1 and pUb in the mouse hippocampus and evaluated protein aggregation at 30- and 70-days post-transfection. At 30 days following sPINK1* transfection, a distinct band for sPINK1 was observed, accompanied by a significant increase in pUb levels (*Figure 6A*). At 70 days following sPINK1* transfection, three distinct bands could be observed that likely correspond to sPINK1, full-length PINK1, and ubiquitinated PINK1 (*Figure 6B*). The presence of full-length PINK1 indicates mitochondrial injury, and ubiquitinated PINK1 indicates compromised proteasomal activity. The pUb levels were increased by 43.8% and 59.9% compared to the control at 30- and 70-days following sPINK1* transfection, respectively. In comparison, the pUb level remained at the control level with the Ub/S65A mutant co-expressed with sPINK1* (*Figure 6C*). Thus, Ub/S65A counteracts the proteasomal inhibition induced by sPINK1*, thereby promoting the degradation of sPINK1* and prevent pUb accumulation.

Immunofluorescence staining revealed the formation of ubiquitin-positive puncta in hippocampal neurons over-expressing sPINK1*, sPINK1+Ub/S65A, or Ub/S65E (*Figure 6D*). The increase of ubiquitin levels in the soluble protein fraction of the hippocampus of brains from the sPINK1*+Ub/S65A group but not in the other groups likely reflects the antibody detection of over-expressed Ub/S65A (*Figure 6E*). On the other hand, the levels of ubiquitin in the insoluble protein fraction were significantly elevated in the sPINK1*, sPINK1*+Ub/S65A, and Ub/S65E groups (*Figure 6F*). Notably, proteins smaller than 70 kDa accumulated in the sPINK1* and Ub/S65E groups, indicating an inhibition of ubiquitin chain elongation, consistent with our findings obtained in vitro (*Figure 3B*), but not in the sPINK1*+Ub/S65A group. Furthermore, overexpression of Ub/S65E resulted in significant neuronal death and glial proliferation (*Figure 6D*, *Figure 5—figure supplements 2–4*), despite that the overall pUb levels appear unchanged at 70 days post-transfection (*Figure 6C*).

It should be noted that overexpression of sPINK1*, sPINK1*+Ub/S65A, or Ub/S65E had minimal effects on autophagic flux at the animal level, as evidenced by unchanged LC3-II and p62 levels in both soluble and insoluble protein fractions (*Figure 6—figure supplement 1*). This contrasts with the findings in cultured cells (*Figure 4C and D*), and suggests that, mechanistically, protein aggregation driven by elevated pUb levels primarily results from the inhibition of proteasomal degradation in the brains of animal or human with neurodegenerative conditions.

## Elevated pUb levels impair neuronal integrity and cognitive function

At 70 days post-transfection with sPINK1*, we observed protein aggregation, mitochondrial injury, neuroinflammation, and proteome-wide changes (*Figures 5 and 6*, *Figure 5—figure supplements 2–4*). To assess the impact of elevated pUb levels on neuronal integrity and cognitive function, we performed behavioral tests on mice at 70 days post-transfection. Over-expression of sPINK1* or Ub/S65E significantly reduced the number of sniffs toward the novel object in novel object recognition test compared to the control mice or the old object (*Figure 7A and B*). In the fear conditioning test, over-expression of sPINK1* or Ub/S65E also significantly reduced freezing time compared to control mice (*Figure 7C and D*). These findings indicate that both sPINK1* and Ub/S65E overexpression impaired cognitive functions of mice. Moreover, co-expression of Ub/S65A with sPINK1* ameliorated sPINK1*-induced cognitive deficits, suggesting a protective effect against pUb-caused impairments (*Figure 7A–D*).

Consistent with the proteomics analysis (*Figure 5*), Western blot analysis showed decreased levels of Tom20, a mitochondrial marker, in brains over-expressing sPINK1* (*Figure 7E*). Notably, the mitochondrial deficit induced by sPINK1* overexpression could be reversed by co-expressing Ub/S65A (*Figure 7E*). Furthermore, we observed reduced levels of MAP2, a dendritic marker, and PSD95, a postsynaptic density marker, in the hippocampus of mice over-expressing sPINK1* and Ub/S65E. The reductions could be restored by co-expressing Ub/S65A with sPINK1* (*Figure 7F and G*). Together, these findings indicate that sPINK1* overexpression induced neuronal injury originating from the nerve terminals, recapitulating neurodegenerative changes. The protective effects conferred upon Ub/S65A co-expression further support the mechanism that the neuronal injury is primarily mediated by elevated pUb levels.

Despite the absence of obvious changes in NeuN immunofluorescence at 70 days post-transfection of sPINK1* (*Figure 5—figure supplement 2*), we investigated morphological alternations in nerve

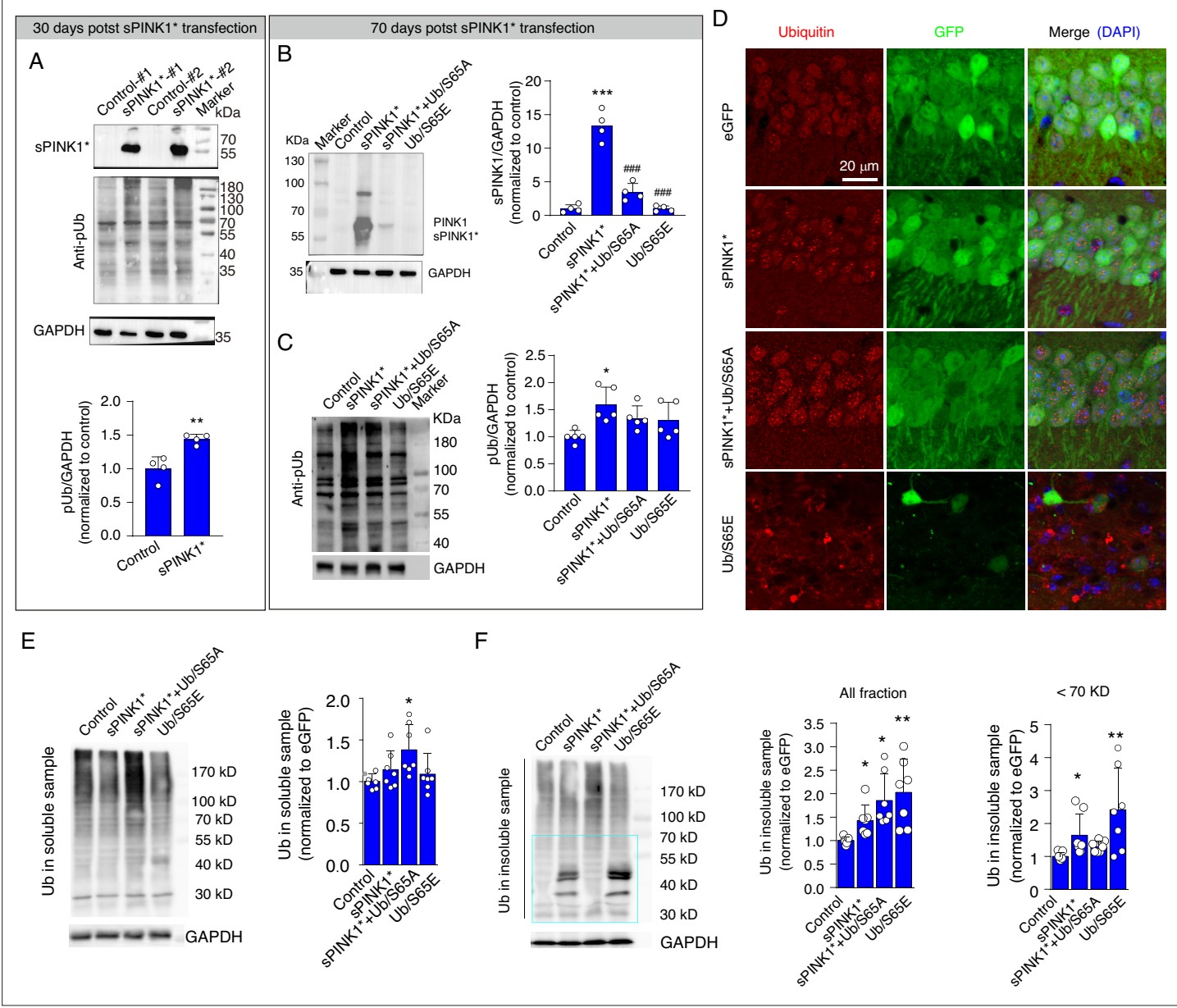

**Figure 6.** Elevated phosphorylated ubiquitin (pUb) levels induce protein aggregation in mouse hippocampal neurons. (**A**) Western blot analysis showing the levels of PINK1 and pUb in mouse hippocampus at 30 days post-transfection. N=4; **p<0.01 compared with control, one-way ANOVA. (**B**) Western blot analysis of PINK1 in the mouse hippocampus at 70 days post-transfection. N=4; **p<0.001 compared with control; ###p<0.001 compared with sPINK1, one-way ANOVA. (**C**) Western blot analysis of pUb in the mouse hippocampus at 70 days post-transfection. N=5; *p<0.05 compared with control, one-way ANOVA. (**D**) Representative immunofluorescence images depicting ubiquitin staining in the CA1 neurons of the mouse hippocampus at 70 days post-transfection. (**E**) Western blot analysis of ubiquitin in the soluble protein fraction of hippocampal lysates at 70 days post-transfection. N=7; *p<0.05 compared with control, one-way ANOVA. (**F**) Western blot analysis of ubiquitin in the insoluble protein fraction of hippocampal lysates at 70 days post-transfection. Quantitative analysis was conducted for total protein at all molecular weight and for proteins with molecular weight <70 kDa. N=7; *p<0.05, **p<0.001 compared with control, one-way ANOVA.

The online version of this article includes the following source data and figure supplement(s) for figure 6:

**Source data 1.** PDF file containing original western blots for **Figure 6A**, indicating the relevant bands and treatments.

**Source data 2.** Original files for western blot analysis displayed in **Figure 6A**.

**Source data 3.** PDF file containing original western blots for **Figure 6B**, indicating the relevant bands and treatments.

**Source data 4.** Original files for western blot analysis displayed in **Figure 6B**.

**Source data 5.** PDF file containing original western blots for **Figure 6C**, indicating the relevant bands and treatments.

*Figure 6 continued*

**Source data 6.** Original files for western blot analysis displayed in *Figure 6C*.

**Source data 7.** PDF file containing original western blots for *Figure 6E*, indicating the relevant bands and treatments.

**Source data 8.** Original files for western blot analysis displayed in *Figure 6E*.

**Source data 9.** PDF file containing original western blots for *Figure 6F*, indicating the relevant bands and treatments.

**Source data 10.** Original files for western blot analysis displayed in *Figure 6F*.

**Figure supplement 1.** Elevated phosphorylated ubiquitin (pUb) level in mouse hippocampus neurons did not perturb the autophagy flow.

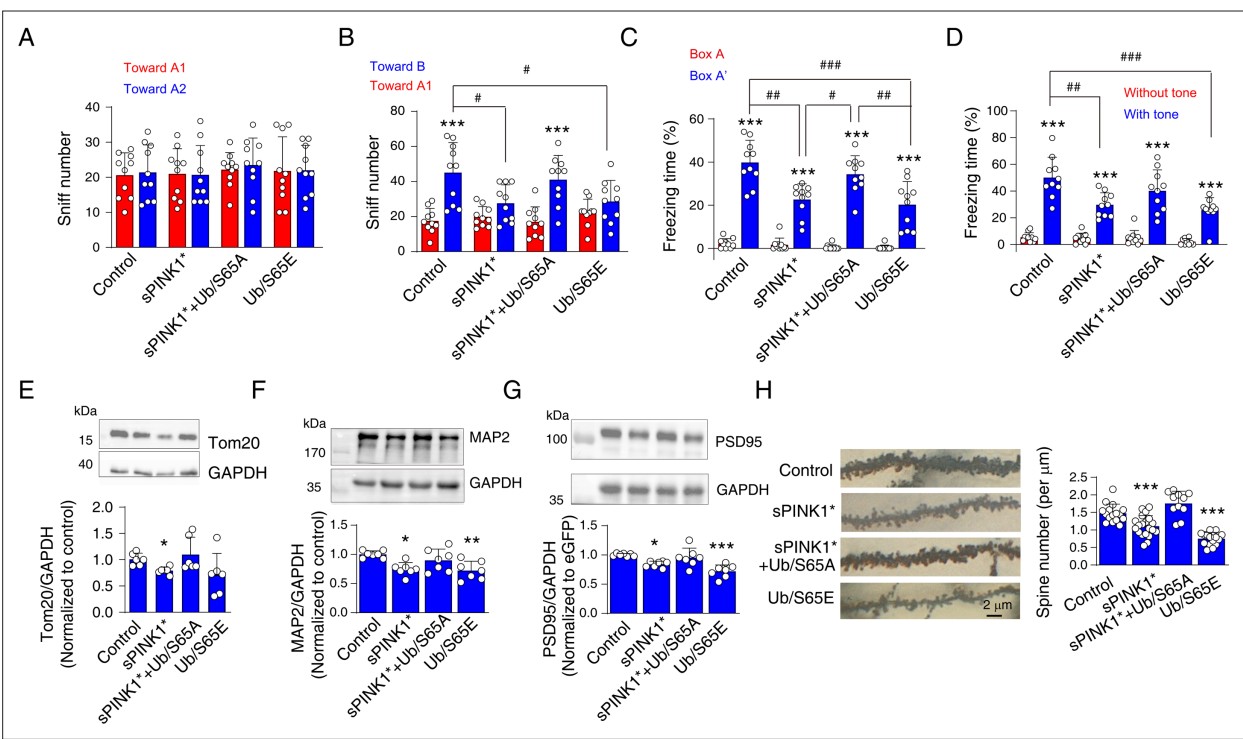

**Figure 7.** Elevated phosphorylated ubiquitin (pUb) levels induce neuronal injury in mouse brains at 70 days post-transfection. (**A**) Novel object recognition test showing the number of sniffs toward two distinct objects in the training phase. N=10. (**B**) Novel object recognition test showing the number of sniffs toward the old object (A1) and the novel object (**B**) in the testing phase. N=10; ***p<0.001 compared with the sniff number toward A1 object, paired t-test. #p<0.05, one-way ANOVA. (**C, D**) The percentage of freezing time in fear conditioning tests for the evaluation of contextual (**C**) and cued (**D**) memory. For contextual memory, the mice were put in the box and received foot shock. For cued memory, the mice were put in a new box, and the same tone accompanying the foot shock was applied. N=10; ***p<0.001 compared with Box A (contextual) or absence of tone (cued), paired t-test. #p<0.05, ##p<0.01, ###p<0.001, one-way ANOVA. (**E–G**) Western blot analyses to assess levels of mitochondrial, dendritic, and synaptic markers: Tom20 (**E**), MAP2 (**F**), and PSD95 (**G**) in the mouse hippocampus. N=7; *p<0.05, **p<0.01, ***p<0.001 compared with control, one-way ANOVA. (**H**) Golgi staining to assess the number of neuronal spines in hippocampal neurons. The left panel shows representative images, and the right panel provides a statistical analysis of spine density on hippocampal neuron dendrites. N=11–20 dendrites from three mice for each group. ***p<0.001 compared with control, one-way ANOVA.

The online version of this article includes the following source data for figure 7:

**Source data 1.** PDF file containing original western blots for *Figure 7E*, indicating the relevant bands and treatments.

**Source data 2.** Original files for western blot analysis displayed in *Figure 7E*.

**Source data 3.** PDF file containing original western blots for *Figure 7F*, indicating the relevant bands and treatments.

**Source data 4.** Original files for western blot analysis displayed in *Figure 7F*.

**Source data 5.** PDF file containing original western blots for *Figure 7G*, indicating the relevant bands and treatments.

**Source data 6.** Original files for western blot analysis displayed in *Figure 7G*.

terminals using Golgi staining. The staining revealed a reduction in dendritic spine density following sPINK1* overexpression, which was reversed by co-expressing Ub/S65A (*Figure 7H*). In contrast, Ub/S65E over-expression resulted in severe dendritic and spine loss, characterized by marked slimmer dendrites and a significant reduction in spine density (*Figure 7H*). These findings are consistent with the Western blot data showing reduced levels of MAP2 and PSD95 (*Figure 7F and G*).

## sPINK1 overexpression inhibits CamKII-CREB pathway in the hippocampus

The inhibition of proteasomal activity not only impairs protein degradation but also disrupts protein activation by preventing the controlled removal of inhibitory proteins. Notably, our proteomics analysis revealed a >54-fold increase in CamK2n1 (Calcium/calmodulin-dependent protein kinase II inhibitor 1, Uniprot Q6QWF9) upon sPINK1* over-expression. CamK2n1 is an endogenous inhibitor of CamKII (*Gouet et al., 2012*), and CamKII is essential for synapse formation, neuronal plasticity, and long-term potentiation (*Hell, 2014*; *O'Day, 2022*). As illustrated in *Figure 8A*, the activation of CamKII requires the targeted degradation of CamK2n1 by the proteasome (*Buard et al., 2010*). Then, the increase of phosphorylated CamKII subsequently initiates the CREB signal pathway, representing a key mechanism in synaptic plasticity (*Davidson and Pickering, 2023*; *Ortega-Martínez, 2015*). The overexpression of sPINK1* increases pUb level, which can inhibit CamK2n1 degradation and suppress the activation of CamKII-CREB signal pathway.

Immunofluorescence staining confirmed the elevated CamK2n1 levels in hippocampus CA1 neurons at 70 days post-transfection of sPINK1* compared to the controls (*Figure 8B*). This increase was accompanied by reduced levels of phosphorylated CamKII (pCamKII) (*Figure 8C*) and phosphorylated CREB (pCREB) (*Figure 8D*), the latter of which was confirmed by Western blotting (*Figure 8E*). Furthermore, sPINK1* over-expression led to reduced levels of BDNF and ERK, all downstream targets of CREB signaling (*Figure 8F and G*). These findings suggest that impaired degradation of CamK2n1 disrupts CamKII-CREB signaling, contributing to the structural deficits in dendritic spines, postsynaptic density, and neuronal projection.

## Discussion

In this study, we demonstrate that elevated pUb levels are a common feature in both chronic and acute neurodegenerative conditions. While mitochondrial injury has been established as a mechanism that causes pUb elevation (*Chin et al., 2023*; *Lambourne et al., 2023*), we have identified an additional pathway—the inhibition of proteasomal degradation leads to the stabilization of sPINK1 in the cytoplasm, where it remains active and phosphorylates ubiquitin.

Proteasomal inhibition can arise from many different pathological factors associated with neurodegeneration (*Cookson et al., 2023*). For example, cerebral ischemia can cause ATP deficiency, and hypoxia and other secondary effect from oxidative stress can further modify the proteasome and impair its function (*Chen et al., 2020*; *Wojcik and Di Napoli, 2004*). In AD, amyloid fibrils can impair proteasomal activities by obstructing the proteasome's substrate entry channel (*McDade et al., 2024*; *Tseng et al., 2008*). Declined proteasomal activity slows the degradation of sPINK1, leading to the elevation of pUb levels in the cytoplasm. Indeed, the elevated pUb levels have been proposed as a biomarker of neurodegeneration (*Fiesel et al., 2015*; *Hou et al., 2018*).

Using both in vitro and in vivo models, we demonstrate that elevated pUb levels, driven by sPINK1* overexpression, inhibit proteasomal activity and exacerbate protein aggregation. Conversely, pink1 knockout reduced protein aggregation in aged mouse brains, brains subjected to MCAO, and HEK293 cells treated with a proteasome inhibitor. Mechanistically, we show that ubiquitin phosphorylation disrupts two critical processes: (1) the covalent elongation of ubiquitin chains, as previously reported (*Wauer et al., 2015b*), and (2) the non-covalent interaction between K48-linked polyUb chains and the proteasome, a novel finding in this study. This dual inhibitory mechanism interferes with the ubiquitination of substrate proteins, the recruitment of ubiquitinated proteins to the proteasome, and ultimately, the proteasomal degradation of substrate proteins. Impaired proteasomal activity further impedes sPINK1 degradation, creating a self-amplifying cycle that drives pUb elevation and progressive proteasomal dysfunction. Although cells have built-in stress responses to perturbation of the ubiquitin-proteasome system (*Hegde et al., 1993*; *Rai et al., 2025*), the disruption caused

by elevated pUb levels targets ubiquitin—the central player in the system—and is self-amplifying as proteasomal function deteriorates.

Extensive literature has reported that pUb plays a neuroprotective role, wherein PINK1 activation in response to mitochondrial damage results in the phosphorylation of ubiquitin and Parkin, initiating mitophagy to remove damaged mitochondria (*Agarwal and Muqit, 2022*; *Li et al., 2023*; *Shiba-Fukushima et al., 2014*). Accordingly, PINK1 deficiency has been linked to neurodegeneration, prompting the development of small-molecule activators (*Chin et al., 2023*; *Hertz et al., 2024*; *Li et al., 2023*). However, in the current study, over-expression of sPINK1 resulted in chronic neuronal injury, driven by the reciprocal relationship between proteasomal inhibition and pUb elevation. The progressive decline of proteasomal activity disrupts cellular proteostasis, leading to the accumulation of protein aggregates. In addition to the mechanism of causing mitochondrial damage and inflammation by the protein aggregates, impaired proteasomal activity hinders the degradation of inhibitory proteins and consequently block the activation of key signaling pathways essential for neurogenesis and memory formation. Collectively, our findings reveal that PINK1 can function beyond the

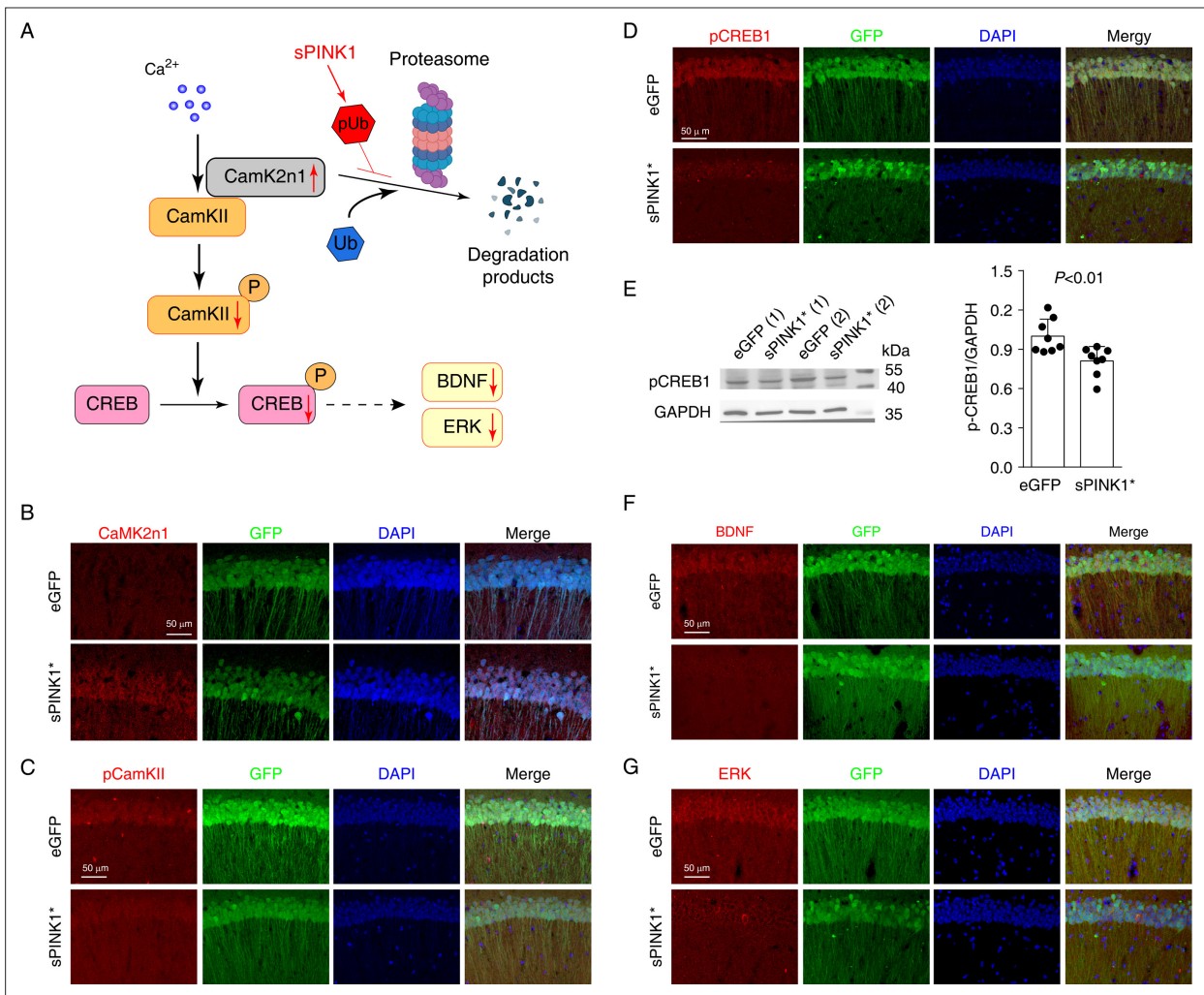

**Figure 8.** Elevated phosphorylated ubiquitin (pUb) levels inhibit CamKII-CREB signaling in mouse hippocampal neurons. (**A**) An illustration of the inhibitory effect on CamK2n1 in the CamKII-CREB1 signaling pathway. (**B–D**) Immunofluorescence staining of CamK2n1 (**B**), pCamKII (**C**), and pCREB1 (**D**) in the CA1 region of hippocampus of mice at 70 days post-transfection. (**E**) Western blot analysis of pCREB1 in mouse hippocampi. N=8, independent t-test. (**F and G**) The immunofluorescence staining of BDNF (**F**) and ERK (**G**) in the CA1 region of mouse hippocampus.

The online version of this article includes the following source data for figure 8:

**Source data 1.** PDF file containing original western blots for *Figure 8E*, indicating the relevant bands and treatments.

**Source data 2.** Original files for western blot analysis displayed in *Figure 8E*.

established role in mitophagy, and elevated pUb levels are not simply a biomarker of neurodegeneration but a driving force behind progressive neuronal injury and neuroinflammation.

The neurotoxic effect upon sPINK1 over-expression is mediated through the ubiquitin phosphorylation. Co-expression of the phospho-null mutant Ub/S65A restored proteasomal activity and even promoted the degradation of the rather stable sPINK1 variant used in the current study. Additionally, Ub/S65A co-expression permitted the formation of longer polyUb chains, further supporting its role in antagonizing ubiquitin phosphorylation. In contrast, over-expression of the phospho-mimetic Ub/S65E mutant caused more severe neuronal damage than sPINK1* overexpression. The difference can be attributed to the 100% increase in pUb mimicry with Ub/S65E, whereas sPINK1*-induced pUb levels are relatively low and subjected to dephosphorylation (*Wall et al., 2019*; *Wang et al., 2018*). Similar toxic effects of Ub/S65E over-expression has been reported in yeast (*Swaney et al., 2015*). Moreover, sPINK1* can phosphorylate both free and substrate-conjugated Ub, whereas Ub/S65E only mimics monomeric pUb, further contributing to the heightened detrimental effects of the Ub/S65E mutant.

In conclusion, our findings have revealed a novel pathogenetic mechanism of neurodegeneration, driven by a self-amplifying cycle of elevated pUb levels and proteasomal inhibition. This cycle leads to the progressive decline of proteasomal activity and the accumulation of toxic protein aggregates, which may underlie both the sequelae of acute insults and the chronic progression of neurodegenerative diseases. Our results demonstrate that pUb is not merely a consequence of this process but also a critical driver of disease progression. Therefore, targeting the pUb-mediated self-amplifying cycle represents a promising therapeutic strategy to slow or halt the progression of neurodegenerative diseases.

## Materials and methods
### Key resources
Key resources are cataloged in the key resources table in the Appendix 1.

### Human brain immunofluorescence staining
The human brain cingulate gyrus paraffin section samples include two samples from people with Alzheimer's Disease and two age-matched samples as the control. The samples with AD were obtained from the Netherlands Brain Bank (NBB), Netherlands Institute for Neuroscience, Amsterdam (Project number 1613), and the age-matched control samples were obtained from the National Health and Disease Human Brain Tissue Resource Center, Hangzhou, China (Project number CN20240454). All material has been collected from donors for whom a written informed consent for a brain autopsy and the use of the material and clinical information for research purposes had been obtained by the NBB and CBB. The detailed information of the donors is provided in *Figure 1—source data 1*. The experiments were conducted with approval from the Ethics Committee of the School of Medicine, Zhejiang University, with the approval number ZJU2023-011.

For immunofluorescence staining, the samples was incubated at 62°C overnight to dewax. The samples were then hydrated—Xylene for three times, 100% ethanol for two times, 95% ethanol for two times, 75% ethanol for two times, and ddH$_2$O for two times. The samples were placed in EDTA antigen retrieval solution, heated to 95–100°C for 30 min, and cooled down at room temperature for 30 min. After washing with PBS buffer for three times, the samples were placed in 3% H$_2$O$_2$ solution for 10 min. After washing with PBS buffer for three times, the samples were incubated in 0.3% Triton-X PBS for 30 min. After washing with PBS buffer for three more times, the samples were permeabilized with 0.3% Tween in PBS for 30 min, and rinsed three times with PBS to remove the detergent. The samples were incubated in a solution with 5% Sudan black B (Aladdin Biochemical Technology Co., Ltd, S109070-25g, Shanghai, China) and 70% ethanol for 1 min at room temperature.

Following PBS rinse, the samples were covered with the blocking solution (5% normal goat serum, 2% bovine serum albumin, 0.1% Tween prepared in TBS-T) for 1 hr. The samples were incubated with rabbit anti-PINK1 antibody (1:200, Novus, Colorado, USA, BC100-494), rabbit anti-pUb antibody (1:200, Millipore, Massachusetts, USA, ABS1513), or mouse anti-Aβ antibody (1:500, Biolegend, California, USA, SIG-39320) at 4°C overnight. After washing with PBS for three times, the samples were incubated with Cy3-conjugated anti-Rabbit IgG (1:200, Jackson, Pennsylvania, USA, 711-165-152) and

fluorescein (FITC)-conjugated anti-mouse IgG (1:200, Jackson, Pennsylvania, USA, 715-096-150) for 2 hr at room temperature. After washing with PBS for three times, the samples were mounted on the slides using a ProLong Gold Antifade Mountant with DAPI (Invitrogen Corp., Carlsbad, CA, USA). The immunofluorescence images were taken using an Olympus FV100 confocal microscope.

## Animals

The C57BL/6J and APP/PS1 mice were purchased from Zhejiang Academy of Medical Science. All mice were kept in the Laboratory Animal Center, Zhejiang University School of Medicine. The mice had free access to water and food in air-conditioned rooms (~26°C, relative humidity ~50%) on a 12-hr light/dark cycle. Mice were handled following the Guide for the Care and Use of the Laboratory Animals of the National Institutes of Health. The experimental protocols were approved by the Ethics Committee of Laboratory Animal Care and Welfare, Zhejiang University School of Medicine, with the approval number ZJU20190138.

Using CRISPR-Cas9-mediated genome editing technology (*Cong et al., 2013*), the Pink1 gene knockout mice (C57BL/6J) were custom-purchased in the Transgenic Mouse Laboratory of the Laboratory Animal Center of Zhejiang University. Briefly, two sgRNA sequences that target exon 6 of the Pink1 gene were cloned into pX330-U6-Chimeric_BB-CBh-hSpCas9 plasmid a gift from Feng Zhang (Addgene plasmid # 42230; http://n2t.net/addgene: 42230; RRID:Addgene 42230) using the Bbsl restriction site. The sequence of sgRNA1 is 5'-GACAGCCATCTGCAGAGAGG-3', and the sequence of sgRNA2 is 5'-GCAGGCAGGACTCACCTCAG-3'. The knockout of pink1 gene was confirmed by using DNA genotype, quantitative PCR, and Western blotting.

## Mouse hippocampus CA1 injection of Adeno-associated virus (AAV)

The rAAV-EF1a-WPRE-hGH pA (AAV2/9) was used to specifically express proteins within neurons, a bistronic vector expressing GFP and sPINK1* (PINK1/F101M/L102-L581) as two separate proteins. Moreover, the rAAV-EF1a-Ub/S65A-P2A-sPINK1-P2A-EGFP-WPRE-hGH pA can express GFP, sPINK1*, and Ub/S65A as three separate proteins, and rAAV-EF1a-Ub/S65E-P2A-EGFP-WPRE-hGH pA can express GFP and Ub/S65E as two separate proteins.

Mice of 9-months-old were anesthetized with pentobarbital sodium (80 mg/kg) by intraperitoneal injection (I.p.). Then the mice were fixed on a mouse stereotaxic holder (RWD Life Science Co. LTP, China) for the injection of AAV. The distinct AAV was stereotactically injected bilaterally into the CA1 region using a microinjector (Hamilton syringe) with a 31-gauge needle and micropump (LEGATO 130 syringe pump, KD Scientific). After anesthesia using pentobarbital sodium (100 mg/kg, i.p.), the injections were made at stereotaxic coordinates of Bregma: anterioposterior (AP) = −1.7 mm, mediolateral (ML)=1.5 mm, dorsoventral (DV)=1.7 mm. A total of 0.3 μl AAV ($2.25 \times 10^{12}$ viral particles/ml) was injected into each side of the hippocampus.

## Behavioral tests

Novel object recognition (NOR) was used to evaluate the spatial memory of mice according to the previous report (*Shen et al., 2023*). After handling for 5 days, the test was performed on the 68th day post-injection. An open-field test system (ViewPoint Behavior Technology, France) was used for NOR detection. During the training, two identical objects (named object A1 and A2, 5 cm' 5 cm' 5 cm blue cone) was placed diagonally. The mouse was placed at the center of the box, and the movement of the mouse was recorded for 10 min. After returning to the home cage for 24 hr, the mouse was put in the same box with one of the objects replaced by a novel object in distinct color and shape (object B, 5 cm' 5 cm' 10 cm yellow cuboid). The sniff number (number of visits) toward the objects was analyzed.

The fear conditioning test was performed in the Fear Conditioning System (Coulbourn Instruments, USA) according to the previous report (*Shen et al., 2023*). The freezing percentage was recorded during each phase to analyze their memory. During the training phase, the mouse was placed in the box (identified as box A) without any stimulation for 1.5 min. Then, a tone (3000 Hz, 85 db) was applied for 30 s, and an electric shock (1 mA) was applied at the final 2 s. After the shock, the mouse was kept in box A for 30 s, and was put back to the home cage. The testing was performed at 24 hr after training. The mouse was placed back in the box A without any stimulation for 5 min, while the freezing percentage in box A indicates contextual memory. Two hours after the test in box A, the mouse was placed in a new box (identified to box B). For the first 2 min, no stimulus was given, and subsequently,

the same tone (3000 Hz, 85 db) was applied for the following 3 min. The freezing percentage in box B without and with tone was used to analyze the cue memory of the mice.

## Focal cerebral ischaemia/reperfusion

After being anesthetized with pentobarbital sodium (80 mg/kg) by intraperitoneal injection (i.p.), the mice were operated with middle cerebral artery occlusion (MCAO) for 2 hr ischaemia followed by 24 hr reperfusion. Briefly, a 6-0 nylon monofilament suture was inserted into the internal carotid to occlude the origin of MCA (*Shi et al., 2012*). Two hours after occlusion, the suture was withdrawn to allow reperfusion for 24 hr. Mouse body temperature was maintained at 37°C using a thermostatic pad during and after surgery. In sham-operated mice, the external carotid artery was surgically isolated but the suture was not inserted.

## Mouse brain tissue collection

Mice were anesthetized using pentobarbital sodium (150 mg/kg, i.p.). The mouse brains were removed after transcardially perfused with 4°C saline and freshly prepared 4% paraformaldehyde. The brains were further fixed in 4% paraformaldehyde at 4°C for one day and transferred to 30% sucrose solution for dehydration. The brains were sliced into 30 μm thick slices by cryomicrotomy (CM1900, Leica, Wetzlar, Germany). The slices were stored at -20°C in the solution (30% glycerol, 30% glycol, 40% PBS) for immunofluorescence and Nissl staining.

For Western blotting, the mouse was transcardially perfused only with 4°C saline. The brains were removed, and the hippocampus was isolated. The hippocampus was quickly frozen in liquid nitrogen and stored at -80°C until use.

## Cell lines, cell culture, and cell transfection

HEK293 cells and SH-SY5Y cells were purchased from the Cell Biology of the Chinese Academy of Sciences (National Collection of Authenticated Cell Cultures, SCSP-5500 and TCHu97, respectively). The PINK1 gene knockout HEK293 cell was homely made using CRISPR-Cas9-mediated genome editing technology (*Ran et al., 2013*). PINK1 sgRNA (CCTCATCGAGGAAAAACAGG) was cloned to U6-sgPINK1-mCherry plasmid a gift from John Doench & David Root (Addgene plasmid # 78038; http://n2t.net/addgene:78038; RRID:Addgene_78038). The U6-sgPINK1-mCherry plasmid and PX458-Cas9-EGFP plasmid (a gift from Prof. Feng Zhang (Broad Institute, Cambridge, MA), Addgene plasmid #48138) were co-transfected into HEK293 cells. The mCherry and GFP double-positive cells were single-cell sorted 24 hr post-transfection using an MoFlo Astrios EQ cell sorter (Beckman Coulter, US) and grown in separate cultures that were subsequently screened for the presence of frameshift mutations leading to nonsense-mediated decay on both alleles. The PINK1 knockout was confirmed using western blotting.

Cells were grown in Dulbecco's modified essential medium (DMEM, Gibco by Thermo Fisher Scientific, C11995500BT) supplemented with 10% heat-inactivated fetal bovine serum (FBS, Zhejiang Tianhang Biotechnology, China, 11011-8611) in the atmosphere of 5% $CO_2$ at 37°C.

Lipo3000 (Invitrogen, California, USA, L3000001) was used to transfect plasmids, following the manufacturer' protocol. Briefly, an appropriate number of cells were seeded one day before transfection with 70-80% confluent at the time of transfection. Before transfection, plasmid DNA was carefully mixed with 250 μl Opti-MEM (Gibco by Thermo Fisher Scientific, Waltham, USA, 31985062). Subsequently, 250 μl Lipo3000 was carefully mixed with Opti-MEM. The mixture was incubated at room temperature for 25 min. The freshly formed DNA/Lipo3000 precipitates were carefully pipetted to the cells. The medium containing transfection reagents was removed after 6 hr and fresh medium was added.

At the end of treatments, the cells on cover slides were washed with 37°C PBS and then fixed using freshly prepared 4% paraformaldehyde. Subsequently, the cells were stored in PBS under 4°C for immunofluorescence staining.

## Oxygen glucose deprivation and reperfusion (OGD/R)

Cells were washed twice by using glucose-free DMEM (Procell, PM150270) and incubated at 37°C with the ventilation of 95% $N_2$ and 5% $CO_2$ for 2 hr. The control cells were washed twice by using high glucose DMEM, and incubated in 95% air and 5% $CO_2$ under 37°C for 2 hr. Then the cells were

cultured in the high glucose DMEM with 10% FBS in 95% air and 5% $CO_2$ under 37°C. This was identified as reperfusion.

## The construction of Ub-R-GFP plasmid

The Ub-R-GFP is used for the in vivo measurement of proteasomal degradation activity, and constructed according to a previous report (*Dantuma et al., 2000*). Briefly, the ubiquitin open reading frame was amplified by PCR with the following primers:

> Sense primer: 5'-GCGGAATTCACCATGCAGATCTTCGTGAAGACT-3'
> Antisense primer: 5'-GCGGGATCCTGTCGACCAAGCTTCCCGCGCCCACCTCTGAG ACGGAGTAC-3'

The PCR product was cloned into the EcoRI and BamHI sites of the EGFP-N2 vector. Besides the R residue, a 12 amino acid peptide was inserted between ubiquitin and GFP to increase the proteasomal degradation. Thus, the final product is Ub-R-GKLGRQDPPVAT-GFP. The accumulation of GFP protein was determined using Western blot, reflecting proteasomal degradation activity.

## Nissl staining

The brain slices were incubated in a mixture of acetone and chloroform (1:1) for 15 min, and then sequentially incubated in 100%, 95%, 70% alcohol for 5 min. The slices were stained for 10 min in Nissl staining containing 0.2% purple crystal (Yuanhang Reagent Factory, Shanghai, China, YHSJ-01–92) and 0.3% acetic acid. Subsequently, the slices were dehydrated by incubation in 70%, 95%, and 100% alcohol. Finally, the slices were put into xylene for 5 min and then mounted with a mixture of neutral resin and xylene (1:1). The images were taken under a microscope (Olympus BX51, Japan).

## Immunofluorescence staining

The floating brain slices or cells on the cover glasses were incubated in 0.1% Triton-X PBS for 30 min, and followed by the incubation in 5% donkey serum for 1 hr. The samples were then incubated with mouse anti-ubiquitin antibody (1:200, Santa Cruz Biotechnology, Texas, USA; SC-8017), rabbit anti-pUb antibody (1:200, Millipore, Massachusetts, USA; ABS1513), rabbit anti-PINK1 antibody (1:200, Novus, Colorado, USA; BC100-494), mouse anti-GFAP antibody (1:1000, CST, Pennsylvania, USA; #3670), rabbit anti-Iba1 antibody (1:1000, FUJIFILM Wako Pure Chemical Corporation, Tokyo, Japan; 019-19741), rabbit anti-NeuN antibody (1:1000, CST, Pennsylvania, USA; #36662), mouse anti-Aβ (1:500, Biolegend, California, USA, SIG-39320), rabbit anti-pCREB antibody (1:1000, ABclonal Technology, Wuhan, China; AP0019), rabbit anti-ERK1/2 antibody (1:200, ABclonal Technology, Wuhan, China; E-AB-31374), rabbit anti-pCamKII antibody (1:200, ABclonal Technology, Wuhan, China; AP0255), rabbit anti-CamK2n1 (1:200, Thermo Fisher, Massachusetts, USA; PA5-23740), and rabbit anti-BDNF antibody (1:200, ABclonal Technology, Wuhan, China; A11028) at 4°C overnight. After washed with PBS (10 min' 3 times), the samples were incubated with Cy3 AffiniPure Donkey Anti-Mouse IgG (H+L) (1:200, Jackson ImmunoResearch, PA, USA; 715-605-150) or Cy3 AffiniPure Goat Anti-Rabbit IgG (H+L) (1:200, Jackson ImmunoResearch, PA, USA; 715-605-150; 111-165-003) or Alexa Fluor488 AffiniPure Goat Anti-Rabbit IgG (H+L) (1:200, Jackson ImmunoResearch, PA, USA; 111-545-003) or Alexa Fluor488 AffiniPure Goat Anti-Rabbit IgG (H+L) (1:200, Jackson ImmunoResearch, PA, USA; 715-545-150) for 2 h. After washing with PBS (10 min each for three times), the slices were mounted on slides using a ProLong Gold Antifade Mountant with DAPI (). The images were taken under an Olympus FV100 confocal microscope (Olympus, Japan) and analyzed using MetaMorph (version 7.8.0.0, Molecular Devices, LLC. San Jose, CA 95134 USA).

## The preparation of soluble and insoluble protein fractions for Western blot analysis

We prepared the soluble and insoluble protein fractions as described in a previous report (*Wirths, 2017*). The cells and brain tissues for the Western blot were lysed using RIPA lysis buffer (Beyotime Biotechnology Research Institute, Jiangsu, China; P0013B) with protease inhibitor (Beyotime Biotechnology Research Institute, Jiangsu, China; P1005) and phosphatase inhibitor (Beyotime Biotechnology Research Institute, Jiangsu, China; P1081). The cells were lysed on ice for 30 min, and every 10 min,

the cells were gently vortexed. The brain samples were thoroughly homogenized with a precooled TissuePrep instrument (TP-24, Gering Instrument Company, Tianjin, China) for 1 min at 4°C. Subsequently, the lysate was centrifuged at 12,000 rpm for 30 min at 4°C. The supernatant was collected as the soluble protein fraction for Western blot analysis.

The precipitate was resuspended in 20 μl SDS buffer (2% SDS, 50 mM Tris-HCl, pH 7.5) for ultrasonic pyrolysis at 4°C. The ultrasonic pyrolysis cycle included 10 s ultrasonic pyrolysis (Diagenode, Seraing, Belgium) and 30 s interval, for a total of eight cycles. The samples were then centrifuged at 12,000 rpm for 30 min at 4°C. The supernatant was collected as the insoluble protein fraction for Western blot analysis.

All protein concentration was determined by using BCA Protein Assay Kit according to the instructions (Beyotime Biotechnology Research Institute, Shanghai, China; P0009).

## Western blot analysis

Appropriate protein samples (50–100 μg) were used and separated using SDS-PAGE gel. The separated proteins were transferred to nitrocellulose membrane. The following antibodies were used: mouse anti-ubiquitin antibody (1:800, Santa Cruz Biotechnology, Texas, USA; SC-8017), rabbit anti-pUb antibody (1:1000, Millipore, Massachusetts, USA; ABS1513), rabbit anti-PINK1 antibody (1:1000, Novus, Colorado, USA; BC100-494), mouse anti-GFAP antibody (1:1000, CST, Pennsylvania, USA; 3670), rabbit anti-CD11b antibody (1:1000, Abcam, Cambridge, UK; ab133357), mouse anti-GAPDH antibody (1:5000, Proteintech, Wuhan, China; 60004–1-Ig), mouse anti-MAP2 antibody (1:2000, Millipore, Billerica, MA, USA; AB5622), rabbit anti-PSD95 antibody (1:1000, CST, Pennsylvania, USA; 3450S), rabbit anti-Tom20 antibody (1:1000, CST, Pennsylvania, USA; 42406), rabbit anti-LC3 antibody (1:1000, Sigma, Massachusetts, USA; L7543), rabbit anti-p62 antibody (1:1000, Abcam, California, USA; ab109012), rabbit anti-pCREB antibody (1:1000, ABclonal Technology, Wuhan, China; AP0019), rabbit anti-GFP antibody (1:1000, Abcam, Cambridge, UK; ab183735), and mouse anti-FLAG antibody (1:10000, TransGen Biotech, Beijing, China; HT201-01). The secondary antibody was HRP-conjugated goat anti-mouse IgG (1:3000, CST, MA, USA; 7076S) or HRP-conjugated goat anti-rabbit IgG (1:10000, Jackson ImmunoResearch, PA, USA, 111-035-003).

The immunoblots were then detected using ECL reagents (Potent ECL kit, MultiSciences Biotech, Hangzhou, China; P1425) and measured using GBOX (LI-COR, Odyssey-SA-GBOX, NE, USA). The results were normalized to GAPDH or Ponceau staining solution (Beyotime Biotechnology Research Institute, Jiangsu, China; P0022), or to the eGFP-expression control on the same immunoblot membrane.

## Golgi staining

On the 70th-day post-transfection, mice were anesthetized by intraperitoneal injection of pentobarbital sodium (150 mg/kg) before sacrifice. The mouse brain was quickly removed, and the dorsal hippocampus was separated. The hippocampus was immediately fixed in the fixative solution (Servicebio, Wuhan, China, G1101) for over 48 h.

The dorsal hippocampus was sliced with a thickness of 2-3 mm thickness around the injection site. The tissue was gently rinsed with PBS at least 3 times. The hippocampus tissue was then placed in Golgi-Cox staining solution (Servicebio, Wuhan, China; G1069), and incubated in a dark for 14 days. The Golgi-Cox staining solution was changed 48 h after the first soak, and every 3 days afterwards. On the 14th-day after staining, the tissues were immersed in distilled water for three times and then incubated in 80% acetic acid overnight until the tissue became soft. After rinsing with distilled water, the tissue was placed into 30% sucrose.

The tissue was cut into 100 μm slices with an oscillating microtome, and the slices were placed on a gelatin slide and dried overnight in the dark. The slices were then treated with ammonia water for 15 min. After washing with distilled water, the slices were incubated in the fixing solution for 15 min. After washing, the slices were sealed using glycerin gelatin. Images were taken using VS120 Virtual Slide Microscope (Olympus, Japan).

## The in vitro ubiquitin-dependent proteasome degradation

Ubiquitin was prepared as previously described (*Liu et al., 2015*; *Liu et al., 2012*). K48-linked di-ubiquitin and tetra-ubiquitin were prepared following an established protocol (*Pickart and Raasi, 2005*),

with the conjugation reaction catalyzed by 2.5 µM ubiquitin-activating enzyme E1 (Uniprot P22314) and 20 µM ubiquitin-conjugating enzyme E2-25K (Uniprot P61086), in 20 mM pH 8.0 Tris-HCl buffer. Ub/K48R mutant was incorporated as the distant Ub, and Ub/77D mutant was incorporated as the proximal ubiquitin to prepare ubiquitin chain of the desired length. The residue D77 at the C-terminus was removed with hydrolase YUH1. The fusion protein His-TEV-Ub-GFP was prepared recombinantly based on the design previously described (*Dantuma et al., 2000*; *Inobe et al., 2011*). The Pediculus humanus corporis PINK1 kinase (phPINK1, uniprot E0W1I1) was prepared and purified as previously described (*Dong et al., 2017*; *Wauer et al., 2015a*). Ubiquitin phosphorylation was confirmed with electrospray mass spectrometry (Agilent G6530 Q-TOF).

The 26S proteasome from HEK293 cells was prepared following an established protocol, with the purification tag appended at the C-terminus of Rpn11 (*Guerrero et al., 2006*; *Huang et al., 2016*). The proteasome activity was assessed with a fluorogenic peptide, N-succinyl-Leu-Leu-Val-Tyr-7-amido-4-methyl coumarin (Sigma, Massachusetts, USA; S6510). A final concentration of 100 nM proteasome and 100 µM proteasome peptide substrate was prepared in the reaction buffer, containing 50 mM Tris-HCl pH 7.5, 4 mM ATP (Sigma- Aldrich, Cat# A6559), and 5 mM MgCl$_2$. The fluorescence intensity of the product was measured by a fluorometer (Horiba Scientific, FluoroMax-4) at an excitation wavelength of 360 nm.

For Western blot analysis of GFP, 200 nM proteasomal substrate and 10 nM human 26 S proteasome were prepared in 20 mM Tris pH 8.0 buffer, with 50 mM NaCl, 5 mM ATP, and 5 mM MgCl$_2$. The mixture was incubated at 37° C for 0, 10, 30, 60, 90, and 120 min. At each time point, 20 µL samples were taken for Western blot analysis. A mouse anti-GFP antibody (1:2000, San-Ying Protein-tech Group, 66002-1) and an HRP-conjugated goat anti-mouse IgG (1:3000, CST, Pennsylvania, USA; 7076S) were used. ImageJ was used to analyze the band intensities, which are normalized to the initial intensity.

The recombinant Ub, pUb (phosphorylated by phPINK1, uniport E0W1I1), Ub/S65A, and Ub/S65E were reacted with 2.5 µM human ubiquitin-activating enzyme E1 and 20 µM ubiquitin-conjugating enzyme 25K in a reaction buffer containing 20 mM Tris-HCl, pH 8.0. The reaction was carried out under standard conditions to promote K48-linked polyubiquitin chain formation. The resulting poly-ubiquitin chains were analyzed by SDS-PAGE, followed by Coomassie blue staining to visualize the chain formation.

## TIRF analysis of ubiquitin-proteasome interaction

The coverslips were prepared following an established protocol (*Roy et al., 2008*). Each coverslip was divided into multiple lanes with double-sided tape for parallel detection of the transient association of K48-polyUb-GFP or pK48-polyUb-GFP in the TIRF experiments, thus to ensure repeatability with different proteins added to the same slide. Streptavidin (VWR Life Science, 97062-808) in imaging buffer at a concentration of 50 µg/ml was loaded; excess unbound streptavidin was washed away. The imaging buffer contains 25 mM Tris-HCl (pH 7.5), 50 mM NaCl, 10 mM MgCl2, 40 mM imidazole, 5 m/mL BSA (Sigma-Aldrich), 2.5 mM ADP (Sigma-Aldrich), and 0.5 mM ATP-γ-S (Sigma-Aldrich), as previously described (*Lu et al., 2015*).

The purified human 26S proteasome at 20 nM in the imaging buffer was added to the streptavidin-immobilized coverslip, with excess unbound proteins washed away with the imaging buffer. The substrate protein, either K48-polyUb-GFP or pK48-polyUb-GFP, was then added to the coverslip to a final concentration of 400 pM. The imaging was performed on a Nikon A1 TIRF microscope equipped with a 488 nm laser and an EMCCD camera (Andor DU-897). The time series was acquired at 300ms per frame for a total duration of 2 min for each time series. The view area was 512 pixels by 512 pixels (16 by 16 µm), and the experiments were repeated four times on four different proteasome-immobilized cover slides. Only when K48-polyUb-GFP or pK48-polyUb-GFP is associated with the immobilized proteasome, the green puncta can be observed. The puncta density was counted in iSMS software (*Preus et al., 2015*) with default settings. The time series of puncta with brightness above a certain threshold (≥ 3000 a.u.) was captured and the dwell time of more than three consecutive frames was selected for counting the overall number of puncta. The dwell times were tabulated in 300 ms bins and fitted with a single exponential decay.

## Proteomics analysis

Hippocampus samples were rinsed using PBS and lysed in RIPA lysis buffer. The samples were then denatured at 95°C for 5 min, followed by sonication at 4°C (3 s on followed with 3 s off, with 30%

amplitude). The lysate was centrifuged at 16,000 g for 10 min at 4°C, and the supernatant was collected as whole tissue extract. Protein concentration was determined by Bradford protein assay. Extracts from the protein sample (2 g) were trypsin digested.

The enriched peptides sample were precipitated in solution A (0.1% formic acid), and was centrifuged at 16,000 g for 10 min. The peptides were separated on a reverse-phase nano-HPLC C18 column (Precolumn, 3 µm, 120 Å, 2x100 i.d.; analysis column, 1.9 µm 120 Å, 30 cm ×150 µm, i.d.) at a flow rate of 600 nL/min with a 150 min gradient of 7-95% solution B (0.1% formic acid in acetonitrile). For peptide ionization, 2100 V was applied, and a 320°C capillary temperature was used. For detection with Q Exactive HF, peptides were analyzed with one full scan (350–1400 m/z, $R = 120,000$ at 200 m/z) with an automatic gain control target of $3 \times 10^6$ ions, with max injection time of 80 ms, followed by up to 30 data-dependent MS/MS scans with high-energy collision dissociation (target $5 \times 10^4$ ions, max injection time 19 ms, isolation window 1.6 m/z, normalized collision energy of 27%), detected in the Orbitrap ($R = 15,000$ at 200 m/z). Dynamic exclusion time was set as 30 s.

Raw MS data were searched against the mouse National Center for Biotechnology Information (NCBI) Refseq protein database (updated on 2013/07/01, 29764 entries) by the software Thermo Proteome Discoverer 2.1 using Mascot 2.3. The mass tolerances were 20 ppm for precursor and 50 mmu for product ions for Q Exactive HF. The search engine set Acetyl(Protein N-term), oxidation(M), Phospho(ST), Phospho(Y) as variable modifications. Trypsin digestion of up to two missed cleavages was allowed. The peptide identifications were accepted at a false discovery rate (FDR) of 1%. Using a label-free approach, a unique peptide was used to represent the absolute abundance of a particular peptide across the sample.

For the proteomic data analysis, quantile normalization was applied to eliminate batch effect, and missing values were imputed using the minimum value observed in the dataset. The differentially expressed proteins were defined with the following criteria: an increase by more than twofold or decrease by more than 50%. For Gene Set Enrichment Analysis (GSEA) analysis, Gene Ontology (GO) analysis was conducted by GSEApy (version 0.10.5) Python package to evaluate the enriched GO terms. The association of proteins with GO terms was derived from the DAVID database. For Gene Set Enrichment Analysis (GSEA), the ranking metric used was the fold change, and significantly enriched GO terms were filtered with a cutoff p-value of 0.05.

## Statistical analysis

Data are presented as mean ± SD. The number in statistical analysis indicates biological replicates. GraphPad Prism Software (version 6.0, GraphPad Software, San Diego, CA, USA) was used for statistical analysis and plot graphs. The ROUT test was used to identify outliers. The Brown-Forsythe test was used to evaluate the equal variances of the data. D'Agostino-Pearson omnibus normality test was performed to test the normality of the data. If the data pass the normality test and equal variance test, t-test, parametric one-way ANOVA (Tukey multiple comparisons test), or two-way ANOVA (Newman-Keuls multiple comparisons test) was used. If not, nonparametric test (Dunn's multiple comparisons test) was used. Chi-square test was also used to assess the differences in categorical data distributions. $p < 0.05$ was considered statistically significant.

## Acknowledgements

This study was supported by the National Key R&D Program of China (2023YFF1204400) to CT and WPZ and (2021YFF1200900) to TTL, the National Natural Science Foundation of China (92353304) to CT and (32070666) to TTL.

## Additional information

### Funding

| Funder | Grant reference number | Author |
| --- | --- | --- |
| National Key Research and Development Program of China | 2023YFF1204400 | Chun Tang Wei-Ping Zhang |

| Funder | Grant reference number | Author |
|---|---|---|
| National Key Research and Development Program of China | 2021YFF1200900 | Tingting Li |
| National Natural Science Foundation of China | 92353304 | Chun Tang |
| National Natural Science Foundation of China | 32070666 | Tingting Li |

The funders had no role in study design, data collection and interpretation, or the decision to submit the work for publication.

### Author contributions

Cong Chen, Data curation, Formal analysis, Methodology, Writing – original draft; Tong-Yao Gao, Data curation, Formal analysis, Writing – original draft; Hua-Wei Yi, Investigation, Visualization, Methodology; Yi Zhang, Software, Formal analysis; Tong Wang, Software, Validation, Methodology; Zhi-Ling Lou, Resources, Methodology; Tao-Feng Wei, Validation, Methodology; Yun-Bi Lu, Supervision, Project administration; Tingting Li, Resources, Software, Supervision; Chun Tang, Conceptualization, Supervision, Funding acquisition, Visualization, Writing – review and editing; Wei-Ping Zhang, Conceptualization, Formal analysis, Supervision, Investigation, Writing – original draft, Writing – review and editing

### Author ORCIDs

Chun Tang ⓘ https://orcid.org/0000-0001-6477-6500
Wei-Ping Zhang ⓘ https://orcid.org/0000-0001-5229-5849

### Ethics

The human brain cingulate gyrus paraffin section samples include two samples from people with Alzheimer's Disease and two age-matched samples as the control. The samples with AD were obtained from the Netherlands Brain Bank (NBB), Netherlands Institute for Neuroscience, Amsterdam (Project number 1613), and the age matched control samples were obtained from the National Health and Disease Human Brain Tissue Resource Center, Hangzhou, China (Project number CN20240454). All material has been collected from donors for or from whom a written informed consent for a brain autopsy and the use of the material and clinical information for research purposes had been obtained by the NBB and CBB. The experiments were conducted with approval from the Ethics Committee of the School of Medicine, Zhejiang University, with the approval number ZJU2023-011.

Mice were handled following the Guide for the Care and Use of the Laboratory Animals of the National Institutes of Health. The experimental protocols were approved by the Ethics Committee of Laboratory Animal Care and Welfare, Zhejiang University School of Medicine, with the approval number ZJU20190138.

Reviewer #1 (Public review): https://doi.org/10.7554/eLife.103945.4.sa1
Author response https://doi.org/10.7554/eLife.103945.4.sa2

---

# Additional files

### Supplementary files

MDAR checklist

### Data availability

All data generated or analyzed during this study are included in the manuscript and supporting files. Source data files have been provided for *Figures 1–8*, including numerical data, PDF scans of original western blots with annotations, and the corresponding raw image files. *Figure 5—source data 1* also contains proteomics quantification data.

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

# Appendix 1

## Appendix 1—key resources table

| Reagent type (species) or resource | Designation | Source or reference | Identifiers | Additional information |
|---|---|---|---|---|
| Genetic reagent (*Mus musculus*) | eGFP(rAAV-EF1a-EGFP-WPRE-hGH-pA) | This paper | sPINK1* and GFP | 5.41E+12vg/ml |
| Genetic reagent (*Mus musculus*) | eGFP +sPINK1(rAAV-EF1a-PINK1(102–581) P2A-EGFP-WPRE-hGH pA) | This paper | sPINK1* and GFP | 2.52E+12vg/ml |
| Genetic reagent (*Mus musculus*) | eGFP +sPINK1+Ub/S65A(rAAV-EF1a-UbS65A-P2A-PINK1(102–581) P2A-EGFP-WPRE-hGH pA) | This paper | Ub/S65A, sPINK1, and GFP | 2.05E+12vg/ml |
| Genetic reagent (*Mus musculus*) | eGFP +Ub/S65E(rAAV-EF1a-UbS65E-P2A-EGFP-WPRE-hGH pA) | This paper | Ub/S65E and GFP | 2.16E+12vg/ml |
| Cell line (*Homo sapiens*) | HEK293 cells | National Collection of Authenticated Cell Cultures | SCSP-5500 | |
| Cell line (*Homo sapiens*) | SH-SY5Y | National Collection of Authenticated Cell Cultures | TCHu97 | |
| Antibody | mouse anti-ubiquitin antibody | Santa Cruz Biotechnology | SC-8017 | WB 1:800; IF 1:200 |
| Antibody | rabbit anti-pUb antibody | Millipore | ABS1513 | WB 1:1000; IF 1:200 |
| Antibody | rabbit anti-PINK1 antibody | Novus | BC100-494 | WB 1:1000; IF 1:200 |
| Antibody | mouse anti-GFAP antibody | CST | #3670 | WB 1:1000; IF 1:1000 |
| Antibody | rabbit anti-Iba1 antibody | FUJIFILM Wako Pure Chemical Corporation | 019–19741 | WB 1:1000; IF 1:1000 |
| Antibody | rabbit anti-NeuN antibody | CST | #36662 | WB 1:1000; IF 1:1000 |
| Antibody | mouse anti-Aβ | Biolegend | SIG-39320 | IF 1:500 |
| Antibody | rabbit anti-pCREB antibody | ABclonal Technology | AP0019 | WB 1:1000; IF 1:1000 |
| Antibody | rabbit anti-ERK1/2 antibody | ABclonal Technology | E-AB-31374 | IF 1:200 |
| Antibody | rabbit anti-pCamKII antibody | ABclonal Technology | AP0255 | IF 1:200 |
| Antibody | rabbit anti-CamK2n1 | Thermo Fisher | PA5-23740 | IF 1:200 |
| Antibody | and rabbit anti-BDNF antibody | ABclonal Technology | A11028 | IF 1:200 |
| Antibody | rabbit anti-Tom20 antibody | CST | 42406 S | WB 1:1000 |
| Antibody | rabbit anti-CD11b antibody | Abcam | ab133357 | WB 1:1000 |
| Antibody | mouse anti-GAPDH antibody | Proteintech | 60004–1-Ig | WB 1:5000 |
| Antibody | mouse anti-FLAG antibody | TransGen Biotech | HT201-01 | WB 1:10000 |
| Antibody | mouse anti-MAP2 antibody | Millipore | AB5622 | WB 1:2000 |
| Antibody | rabbit anti-PSD95 antibody | CST | 3450 S | WB 1:1000 |
| Antibody | rabbit anti-GFP antibody | Abcam | ab183735 | WB 1:1000 |
| Antibody | rabbit anti-LC3 antibody | Sigma | L7543 | WB 1:1000 |
| Antibody | rabbit anti-p62 antibody | Abcam | ab109012 | WB 1:1000 |
| Antibody | HRP-conjugated goat anti-rabbit IgG | Jackson ImmunoResearch | 111-035-003 | WB 1:10000 |
| Antibody | HRP-conjugated goat anti-mouse IgG | CST | 7076 S | WB 1:3000 |
| Antibody | Cy3 AffiniPure Donkey Anti-Mouse IgG (H+L) | Jackson ImmunoResearch | 715-605-150 | IF 1:200 |
| Antibody | Cy3 AffiniPure Goat Anti-Rabbit IgG (H+L) | Jackson ImmunoResearch | 111-165-003 | IF 1:200 |
| Antibody | Alexa Fluor488 AffiniPure Goat Anti-Rabbit IgG (H+L) | Jackson ImmunoResearch | 111-545-003 | IF 1:200 |
| Antibody | Alexa Fluor488 AffiniPure Goat Anti-Rabbit IgG (H+L) | Jackson ImmunoResearch | 715-545-150 | IF 1:200 |
| Recombinant DNA reagent | PRK5/sPINK1/IRES/EGFP(plasmid) | This paper | sPINK1* and GFP | |

*Appendix 1 Continued on next page*

*Appendix 1 Continued*

| Reagent type (species) or resource | Designation | Source or reference | Identifiers | Additional information |
|---|---|---|---|---|
| Recombinant DNA reagent | PRK5/EGFP(plasmid) | This paper | GFP | |
| Recombinant DNA reagent | PRK5/UbS65E/IRES/EGFP(plasmid) | This paper | Ub/S65E and GFP | |
| Recombinant DNA reagent | PRK5/sPINK1-P2A-UbS65A/IRES/EGFP(plasmid) | This paper | Ub/S65A, sPINK1, and GFP | |
| Recombinant DNA reagent | FLAG-Ub/48 K(plasmid) | This paper | Ub/48 K | |
| Recombinant DNA reagent | pcDNA3.1/sPINK1(plasmid) | This paper | sPINK1* and GFP | |
| Recombinant DNA reagent | pcDNA3.1/PINK1(plasmid) | This paper | PINK1 | |
| Recombinant DNA reagent | pRK5-sPINK1(K219A,D362A,D384A)-IRES-EGFP(plasmid) | This paper | sPINK1(K219A,D362A,D384A) and GFP | |
| Recombinant DNA reagent | pRK5-PINK1-IRES-EGFP(plasmid) | This paper | PINK1 and GFP | |
| Recombinant DNA reagent | pIRES2-Ub-GG-sPINK1-EGFP(plasmid) | This paper | Ub-GG-sPINK1+EGFP | |
| Recombinant DNA reagent | pEGFP-N2-Ub-R-GFP(plasmid) | This paper | Ub-R-GFP | |
| Commercial assay or kit | Dulbecco's modified essential medium (DMEM) | Gibco by Thermo Fisher Scientific | C11995500BT | |
| Commercial assay or kit | Fetal bovine serum | Zhejiang Tianhang Biotechnology | 11011–8611 | |
| Commercial assay or kit | Fetal bovine serum | Gibco by Thermo Fisher Scientific | 10099–141 | |
| Commercial assay or kit | MEM Non-Essential Amino Acids | Gibco by Thermo Fisher Scientific | 11140–050 | |
| Commercial assay or kit | Lipo3000 | Invitrogen Corp. | L3000001 | |
| Commercial assay or kit | Opti-MEM | Gibco by Thermo Fisher Scientific | 31985062 | |
| Commercial assay or kit | purple crystal | Yuanhang Reagent Factory | YHSJ-01–92 | |
| Commercial assay or kit | ProLong Gold Antifade Mountant with DAPI | Invitrogen Corp. | P36931 | |
| Commercial assay or kit | ECL kit | MultiSciences Biotech | P1425 | |
| Commercial assay or kit | Ponceau staining solution | Beyotime Biotechnology Research Institute | P0022 | |
| Commercial assay or kit | RIPA lysis buffer | Beyotime Biotechnology Research Institute | P0013B | |
| Commercial assay or kit | protease inhibitor | Beyotime Biotechnology Research Institute | P1005 | |
| Commercial assay or kit | phosphatase inhibitor | Beyotime Biotechnology Research Institute | P1081 | |
| Commercial assay or kit | Ponceau staining solution | Beyotime Biotechnology Research Institute | P0022 | |
| Commercial assay or kit | BCA protein assay kit | Beyotime Biotechnology Research Institute | P0009 | |

*Appendix 1 Continued on next page*

*Appendix 1 Continued*

| Reagent type (species) or resource | Designation | Source or reference | Identifiers | Additional information |
|---|---|---|---|---|
| Commercial assay or kit | MG132 | MedChemExpress | HY-13259 | |
| Commercial assay or kit | Puromycin | Thermo Fisher Scientific | A1113802 | |
| Commercial assay or kit | bafilomycin A1 (BALA) | Sigma-Aldrich | 508409 | |
| Commercial assay or kit | Golgi-cox staining solution | Servicebio | G1069 | |
| Commercial assay or kit | Sudan black B | Aladdin Biochemical Technology Co. | S109070-25g, | |
| Commercial assay or kit | protein kinase | Sangon Biotech (Shanghai) Co. | B600452 | |
| Commercial assay or kit | Streptavidin | VWR Life Science | 97062–808 | |

