## [Editor Report · eLife Assessment]

This study provides **important** insights into the role of polyUbiquitination in neurodegenerative diseases, elucidating how pUb promotes neurodegeneration by affecting proteasomal function. The findings not only offer a new perspective on the pathophysiology of neurodegenerative diseases but also provide potential targets for developing new therapeutic strategies. The results provide **solid** evidence to support the conclusions.

---

## [Referee Report · Reviewer #1 (Public review)]

Summary:

The manuscript discusses the role of phosphorylated ubiquitin (pUb) by PINK1 kinase in neurodegenerative diseases. It reveals that elevated levels of pUb are observed in aged human brains and those affected by Parkinson's disease (PD), as well as in Alzheimer's disease (AD), aging, and ischemic injury. The study shows that increased pUb impairs proteasomal degradation, leading to protein aggregation and neurodegeneration. The authors also demonstrate that PINK1 knockout can mitigate protein aggregation in aging and ischemic mouse brains, as well as in cells treated with a proteasome inhibitor. While this study provided some interesting data, several important points should be addressed before being further consideration.

Strengths:

(1) Reveals a novel pathological mechanism of neurodegeneration mediated by pUb, providing a new perspective on understanding neurodegenerative diseases.

(2) The study covers not only a single disease model but also various neurodegenerative diseases such as Alzheimer's disease, aging, and ischemic injury, enhancing the breadth and applicability of the research findings.

Comments on revisions:

This study, through a systematic experimental design, reveals the crucial role of pUb in forming a positive feedback loop by inhibiting proteasome activity in neurodegenerative diseases. The data are comprehensive and highly innovative. However, some of the results are not entirely convincing, particularly the staining results in Figure 1.

In Figure 1A, the density of DAPI staining differs significantly between the control patient and the AD patient, making it difficult to conclusively demonstrate a clear increase in PINK1 in AD patients. Quantitative analysis is needed. In Fig 1C, the PINK1 staining in the mouse brain appears to resemble non-specific staining.

---

## [Author Response]

The following is the authors’ response to the previous reviews

In response to Reviewer #1, we have replaced the original images in Figure 1A with new immunofluorescence data showing matched DAPI staining density between control and AD patient samples. We also have updated the PINK1 staining images of mouse brain sections in Figure 1C to eliminate potential non-specific signals. These revisions provide clearer evidence supporting our conclusions about PINK1/pUb’s role in neurodegeneration.